# Cytoskeletal components can turn wall-less spherical bacteria into kinking helices

Carole Lartigue [1], Bastien Lambert[1], Fabien Rideau [1], Yorick Dahan [1], Marion Decossas [2], Mélanie Hillion [3,4], Jean-Paul Douliez[1], Julie Hardouin[3,4], Olivier Lambert[2], Alain Blanchard [1] & Laure Béven [1] ✉

Bacterial cell shape is generally determined through an interplay between the peptidoglycan cell wall and cytoplasmic filaments made of polymerized MreB. Indeed, some bacteria (e.g., *Mycoplasma*) that lack both a cell wall and *mreB* genes consist of non-motile cells that are spherical or pleomorphic. However, other members of the same class *Mollicutes* (e.g., *Spiroplasma*, also lacking a cell wall) display a helical cell shape and kink-based motility, which is thought to rely on the presence of five MreB isoforms and a specific fibril protein. Here, we show that heterologous expression of *Spiroplasma* fibril and MreB proteins confers helical shape and kinking ability to *Mycoplasma capricolum* cells. Isoform MreB5 is sufficient to confer helicity and kink propagation to mycoplasma cells. Cryoelectron microscopy confirms the association of cytoplasmic MreB filaments with the plasma membrane, suggesting a direct effect on membrane curvature. However, in our experiments, the heterologous expression of MreBs and fibril did not result in efficient motility in culture broth, indicating that additional, unknown *Spiroplasma* components are required for swimming.

Maintenance and dynamic reconfiguration of cell shape represent a selective value for bacteria both for primary and secondary cellular processes, in particular for nutrient acquisition, division, the capacity to escape from predators, biofilm formation, and motility[1]. Therefore, natural evolution has led most bacterial species to adopt one or a limited number of morphologies among many more or less complex possibilities, depending on their way of life and ecological niche[2]. The main determinant of bacterial cell shape is the peptidoglycan layer surrounding the plasma membrane and forming the cell wall[3]. The morphological transition of cells into spheres upon inhibition of cell wall synthesis in rod-shaped bacteria demonstrates the essential role of this structure in the maintenance of an elongated bacterial morphology[4]. In most rod-shaped bacteria, short internal filaments made up of actin-like proteins called MreBs guide the synthesis machinery of the cell wall to ensure cell elongation as deposition and crosslinking of new peptidoglycan units progress[5].

Along with cocci and rods, helical or corkscrew morphologies are major shapes adopted by phylogenetically distant bacteria, including *Helicobacter pylori*, spirochetes and spiroplasmas. In *H. pylori*, the helical shape of the cell body can significantly contribute to propulsive thrust[6] or to pathogenicity[7]. In this species, the cell wall is differentially synthesized based on the curvature of the cell body, with two proteins, MreB and CcmA, defining the appropriate areas where synthesis is enhanced[8]. Helical or wave-like morphologies and motility in spirochetes are primarily determined by the periplasmic flagella, cell wall, and cytoplasmic MreB[9]. Spiroplasmas represent a group of helical bacteria apart. Indeed, spiroplasmas belong to the class Mollicutes, characterized by the lack of a peptidoglycan-based cell wall[10]. Most *Spiroplasma* species are pathogens or endosymbionts of arthropods and plants[11]. With the sole exception of the strain S*piroplasma citri* ASP-1, all natural *Spiroplasma* isolates described to date are helical and motile[12], suggesting a selective value of this specific shape and unique

[1]Univ. Bordeaux, INRAE, BFP, UMR 1332, Villenave d'Ornon, France. [2]University Bordeaux, CNRS, CBMN UMR 5248, Bordeaux INP, Pessac, France. [3]University of Rouen Normandy, UNIROUEN, INSA Rouen, CNRS, Polymers, Biopolymers, Surfaces Laboratory, Rouen, France. [4]University of Rouen Normandy, INSERM US 51, CNRS, UAR 2026, HeRacLeS-PISSARO, Rouen, France. ✉e-mail: laure.beven@inrae.fr

motility. Thus, spiroplasmas control their helicity without a peptidoglycan layer, the major determinant of bacterial shape in walled bacteria. In addition, these bacteria are motile in the absence of external appendages, such as flagella or pili, which allow motility of the vast majority of bacteria[13]. Recently, there have been an increasing number of studies aimed at elucidating the mechanisms of shape maintenance and motility in *Spiroplasma*. Indeed, *Spiroplasma* appears to be a particularly attractive model for the identification of shape- and motility-determining factors in a wall-less organism[12].

*Spiroplasma* cells are polarized, with a tapered (also called tip) and a rounded end[14]. The discovery of an internal cytoskeleton composed of the protein fibril (Fib) unique to *Spiroplasma*[15] occurred quickly after the discovery of these bacteria[16]. The main cytoskeletal structure corresponds to a monolayered, flattened ribbon positioned along the shortest helical path[17]. Microscopic observations of cryofixed freeze-substituted preparations combined with tomographic reconstruction confirmed the overall organization and highlighted the membrane association of the internal protein ribbon consisting of both actin-like MreBs and fibril[18]. The internal cytoskeleton also comprises a dumbbell-shaped structure at one cell pole (tapered-end) that is likely involved in cell polarization[19]. The motility of *Spiroplasma* is due to a helicity change, which is initiated at one of the two ends of the cell and introduces a "kink" in the cell whose propagation is responsible for movement of the bacteria[20]. On the cytoplasmic side, the complex dumbbell-shaped structure could be responsible for the generation of the initial twist, which is then propagated along the cell[21].

The presence of at least 5 *mreB* paralogues in the near-minimal genome of *Spiroplasma* species[22] raises some questions regarding the selective benefit provided by the different isoforms during evolution. In the nonhelical strain *S. citri* ASP-1, the loss of helicity and motility is due to a nonsense mutation within the sequence encoding MreB5 that cannot be functionally compensated by any of the other 4 isoforms present in this species[23]. Thus, MreB5 was identified as a major determinant of cell helicity in *S. citri*[23]. These in vivo results, coupled with differences in polymerization and depolymerization dynamics between isoforms[24–26], strengthen the hypothesis of functional differentiation between some MreB paralogues.

Despite the qualitative compositional information, the structure and mechanisms of the *Spiroplasma* internal cytoskeleton remain unclear. Recently, based on the structure of the Fib filament, a molecular mechanism has been proposed in which fibril is responsible for cell helicity and for its shift. Following this model, the length changes in MreB polymers would generate a helicity shift[21]. The construction of *Spiroplasma* mutants expressing combinatorial sets of MreBs with or without fibril could help validate this motility model and identify the different roles of MreBs. However, although different tools have been developed to modify the *Spiroplasma* genome, it remains difficult to obtain conditional gene knockdowns, especially for paralogues[12]. To circumvent these limitations, one approach is to perform heterologous expression experiments of *Spiroplasma* cytoskeletal proteins within phylogenetically related *Mycoplasma* species also lacking a cell wall. The choice of *Mycoplasma capricolum* subsp. *capricolum* (*Mcap*) is justified as it belongs to the Spiroplasma phylogenetic group of the Mollicutes class, and it is amenable to genome engineering, including the use of methods derived from synthetic biology[27,28]. *Mcap* cells are coccoid, and their genome does not encode any fibril or MreB protein. In addition, *Mcap* and *Spiroplasma* membranes have a similar lipid composition[29,30], which is an essential point when considering that cytoskeletal elements were found to be closely associated with the membrane[18].

Here, we investigated whether reconstruction of the *Spiroplasma* cell structure in *Mcap* was possible. We then took advantage of this model to obtain clues on the minimal requirements for helical shape and kinking motility in Mollicutes by comparing the effect of the insertion of different combinations of *S. citri* genes in the *Mcap*

genome on morphology, motility and the formation of internal cytoskeletal elements. Throughout the following text, the term motility will be used to refer to the ability of the bacteria to move in liquid medium, resulting in an active directional displacement of the cells. The term kink propagation refers to movements of the cell membrane that induce a shift in helicity as they propagate along the cell body.

## Results

### Expression of *Spiroplasma* cytoskeletal proteins in *Mcap* confers helicity and kink propagation

Recombinant strains of *M. capricolum* subsp. *capricolum* (*Mcap* hereafter) with *mreB* and fibril-encoding genes from *S. citri* were obtained after transformation with transposons (Supplementary Fig. 1). As a control, *Mcap* cells were transformed with the transposon vector alone without additional genes (*Mcap*^control). The morphology of the recombinant cells was initially analyzed by darkfield microscopy after 4 passages in SP4 medium. Control cells were pleomorphic as expected for *Mcap* with a mean cell length $L = 2.6 \pm 2.0\,\mu m$ (Fig. 1a), and none showed a helical morphology. In contrast, with the recombinants resulting from transformation with the transposon carrying the combination of *mreB1-5* and fibril genes and named *Mcap*^mreB1-5-fib, the cell morphology of elongated cells ($L > 3\,\mu m$, $n = 150$) was heterogeneous and included helical (49%) and nonhelical filamentous cells (51%) up to 80 microns in length (Fig. 1a–c). A large number (47%) of elongated cells also showed branching (Fig. 1b and Supplementary Fig. 2). Fluorescence-based viability assays indicated that elongated cells, branched or not, were viable (Supplementary Fig. 3). Small viable cells ($L < 3\,\mu m$) were also visible (Fig. 1a and Supplementary Fig. 2). However, due to their small size, it was not possible to identify their exact morphology (coccoid, rod-shaped or curved) using darkfield microscopy (Supplementary Fig. 2). When elongated cells were helical, the helical pitch, corresponding to the distance between two equivalent points separated by a single turn on the helix, could be measured parallel to the cell axis and determined from darkfield microscopy images (insert, Fig. 1c). It was $1.7 \pm 1.0\,\mu m$ on average ($n = 50$), which was significantly different from the $0.70 \pm 0.1\,\mu m$ ($n = 50$) determined for *S. citri* (Fig. 1c). Up to 20% ($n = 50$) of the helical cells exhibited bending and kinking movements, mimicking those seen with *S. citri* (Fig. 2 and Supplementary Movies 1, 2). However, kinks were initiated irregularly but locally traveled with a similar mean velocity $V_{kink}$ in *S. citri* ($V_{kink} = 12.0 \pm 2\,\mu m/s$, $n = 10$) and in *Mcap*^mreB1-5-fib ($V_{kink} = 10.8 \pm 2\,\mu m/s$, $n = 10$). Kink propagation was observed for helical pitches between 0.6 and $1.5\,\mu m$. For comparison, a helical cell with no kinks, only submitted to Brownian motion, can be viewed in Supplementary Movie 2. Erratic movements were also observed for nonhelical filaments, inducing bending of the elongated filaments. Additionally, in SP4 broth, some cell bodies showed certain flexibility, while others were straight and inflexible (the cell was not deformed upon Brownian motion) (Supplementary Movie 2).

The kink-based cell movements did not provide directional motility to helical *Mcap*^mreB1-5-fib cells in SP4 broth. Since the medium viscosity minimizes Brownian motion and favors *S. citri* motility[31], 0.2 to 1% methylcellulose was added to the medium, but this additive did not improve the translational movement of the *Mcap*^mreB1-5-fib recombinant (data not shown). Additionally, while *S. citri* grown on SP4 agar-containing plates formed satellite colonies due to the migration of single cells away from the mother colony, no satellites were observed with the *Mcap*^mreB1-5-fib recombinant after growth on nutrient medium containing 0.8% Noble agar (Supplementary Fig. 4), suggesting that the propulsive force conferred by the addition of *S. citri* cytoskeletal proteins was not sufficient to efficiently move the cells in one direction, at least under the conditions tested.

Expression of MreBs and Fib in *Mcap* had not only a significant effect on cell morphology but also on cell division. Indeed, the presence of long filaments was likely the result of defective septation during the

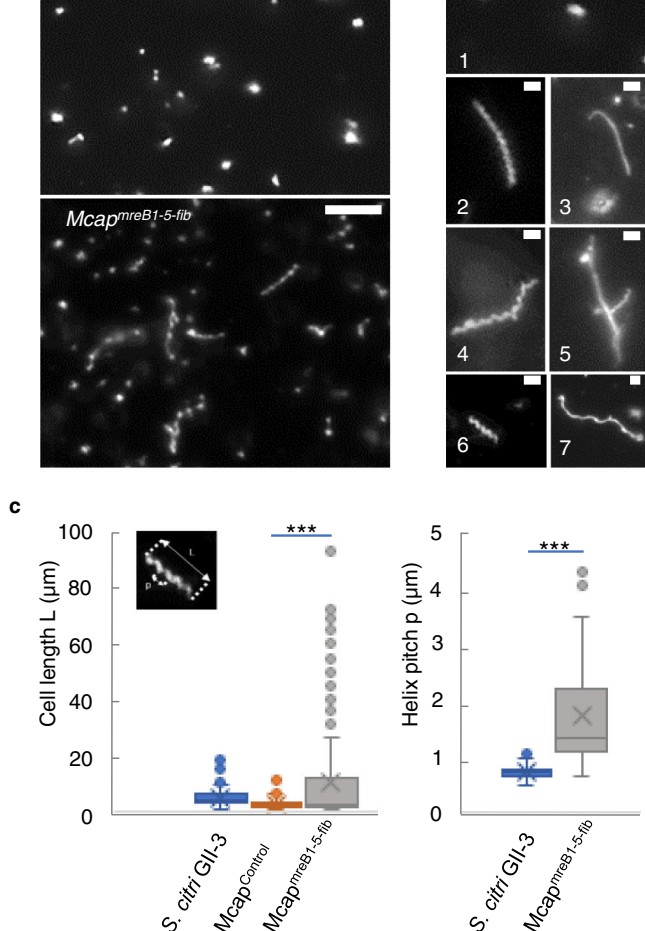

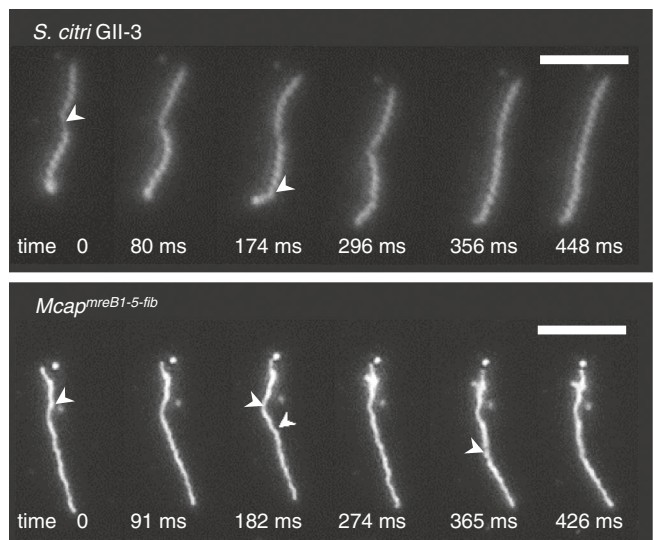

**Fig. 2 | Time-lapse images showing the kink-based cell movements.** Darkfield microscopy images were recorded in *S. citri* (top) and in *Mcap^mreB1-5-fib^* (bottom). White arrows point to kinks. Note the helicity shift upon propagation of the kink along the cell body. These micrographs are representative of at least 3 independent experiments. Scale bar: 5 μm.

**Fig. 1 | Morphology and motility of *Spiroplasma citri* GII-3 and *Mcap* transformants with genome insertions of *mreB* and *fib* genes of *S. citri*. a** Large darkfield images of cultures of *Mcap^control^* and *Mcap^mreB1-5-fib^* cells, imaged area approximately 62 μm wide. Scale bar: 10 μm. These micrographs are representative of at least 3 independent experiments. **b** Darkfield microscopy images showing representative morphologies of short rod-shaped *Mcap^control^* cells (1), and of *Mcap^mreB1-5-fib^* cells: long helical (2) and nonhelical (3) cells; a branched helical cell (4); a straight, branched filament showing helical or straight lateral extensions (5); a short helical cell (6); and a long wavy cell (7). These micrographs are representative of at least 3 independent experiments. Scale bar: 2 μm. **c** Box plot display of the cell length and helical pitch in *S. citri*, *Mcap^control^* transformed with the empty vector and *Mcap^mreB1-5-fib^* cell populations measured in darkfield microscopy images. The central line corresponds to the median. The lower and upper hinges of the boxes correspond to the 25th and 75^th percentiles, and the whiskers represent the 1.5×interquartile range extending from the hinges. Points are outliers. The top left insert represents the helical pitch p and the length L of a representative helical cell. Helical pitch measurements were restricted to helical cells, and were made after 4 passaging in liquid broth. Data were collected for 150 cells (cell length) or 50 cells (helical pitch) from 3 biological replicates for each cell population and compared by the two-tailed Student's t test; \*\*\* indicates a significant difference between populations at $p < 0.001$. Source data are provided as a Source Data file.

## Fibril is not essential for helicity or for kinking capacity but favors the propagation of deformations at the membrane level along the cell body in *Mcap*

We took advantage of the *Mcap* heterologous system to assess whether helicity and motility could be conferred to cells when *fib* only or *mreB1-5* only were inserted in the genome of recombinants (Supplementary Fig. 5). To avoid the possible impact of the localization of the added genes in the *Mcap* genome (Supplementary Table 1), only features common to all clones tested are described below.

After 4 passages in SP4 medium, *Mcap^fib^* recombinants were characterized by a coccoid or short rod shape for most cells but also by the presence of a few short helices (<1% of the cells) having a helical pitch of $1.08 \pm 0.29$ μm (Fig. 3a and Supplementary Fig. 2). Helices were also observed in the cell population of *Mcap^mreB1-5^* recombinants. Among elongated cells with a length >3 μm ($n = 150$), helical *Mcap^mreB1-5^* represented up to 55% of the cells (Supplementary Fig. 2), with a mean pitch of $1.8 \pm 0.7$ μm (Fig. 3a), which was not significantly different from the pitch of *Mcap^mreB1-5-fib^* transformants ($1.7 \pm 1.0$ μm). Thus, MreBs were sufficient to confer helicity to *Mcap* cells. It was noticeable that for both *Mcap^fib^* and *Mcap^mreB1-5^* transformants, the range of helical pitch lengths overlapped with the length of that of *S. citri*, indicating that the *Spiroplasma* shape could be found with either construction. Defects in septation were observed for more than 60% ($n = 150$) of elongated *Mcap^mreB1-5^* cells, with the formation of long, branched helices or nonhelical long filaments. One possible explanation for this finding was that one or several MreBs might interfere with the division process. These division problems were associated with the formation of brownish, irregular colonies on agar plates that were smaller than those observed with the control cells, *i.e., Mcap* transformed with empty vector (Supplementary Fig. 4).

For both *Mcap^mreB1-5^* and *Mcap^fib^* recombinants, cell movements were observed for helices with a helical pitch between 0.7 and 1.5 μm. The addition of both gene sets allowed the propagation of membrane deformations along the cell body and was responsible for a change in helicity in 8% ($n = 50$) of *Mcap^mreB1-5^* helical cells (Fig. 4, Top panel). In *Mcap^fib^* recombinants, the propagation of membrane deformation triggered the transient loss of helicity. Consequently, the helical shape of these recombinants could only be seen during the resting period

cell division process. Elongation of cells that were not correctly separated resulted in branching (Fig. 1b). Impairment of cell division was also correlated with a particular aspect of the colonies grown on SP4-agar medium: while the control transformants gave typical fried-egg colonies, *Mcap^mreB1-5-fib^* recombinant growth resulted in the formation of colonies with irregular contours (Supplementary Fig. 4). Such colony morphology could be associated with division arrest for some cells.

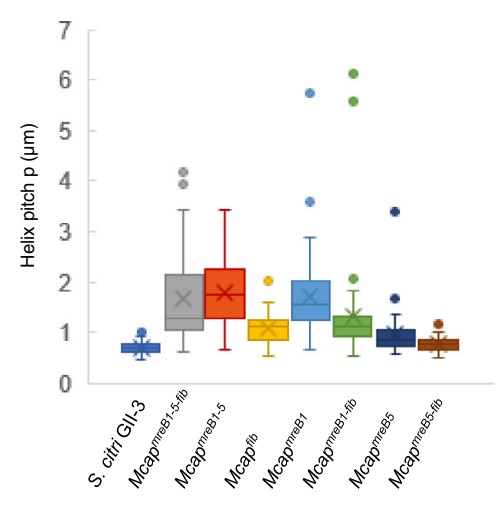

**a**

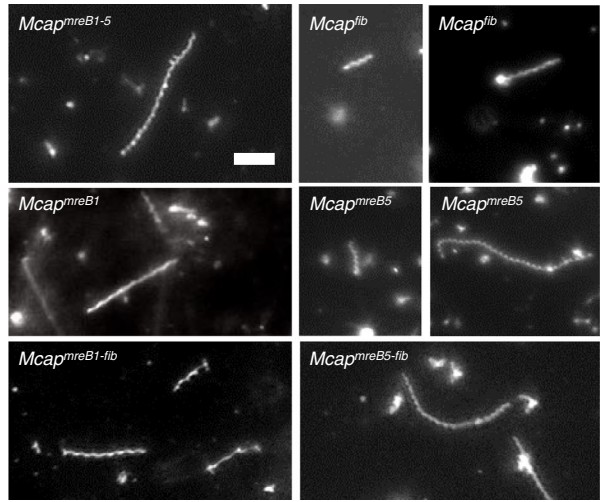

**b**

**Fig. 3 | Helicity of *Mcap* transformants bearing different combinations of *mreB* and *fib* genes observed using darkfield microscopy. a** Box plot display of the helical pitch in *Mcap* transformants. Data were collected for 50 cells from 3 biological replicates for each transformant population. The central line corresponds to the median. The lower and upper hinges of the boxes correspond to the 25th and 75$^{th}$ percentiles, and the whiskers represent the 1.5×interquartile range extending from the hinges. Points are outliers. Source data are provided as a Source Data file. **b** Representative helical cells observed in *Mcap* transformed with the different plasmid constructs. These micrographs are representative of at least 3 independent experiments. Scale bar: 5 μm.

between two propagating deformations (Fig. 4, bottom panel). Thus, MreBs only or Fib only endowed *Mcap* with the possibility of mimicking *Spiroplasma* membrane deformations responsible for its motility. These movements were, however, not sufficient to move the cells in one direction in SP4 liquid (Supplementary Movie 3) or methyl cellulose-supplemented semiviscous medium (data not shown). Kinking helical cells all had a length less than 5 μm. A difference in capacity in propagating membrane deformations between *Mcap*$^{mreB1-5-fib}$ and *Mcap*$^{mreB1-5}$ transformants was noticed: long, nonhelical filaments were observed in both cell populations (Supplementary Fig. 2); however, bending of such filaments was only observed in the case of *Mcap*$^{mreB1-5-fib}$ (Supplementary Movie 2).

Taken together, these results suggested that fibril was not essential for helicity or motility but likely increased the efficiency of propagation of membrane deformation (kink) along the cell body. In addition, Fib by itself could confer helicity and kinking capacity to short helices. Noticeably, subculturing of *Mcap*$^{fib}$ recombinants in

liquid SP4 medium led to the loss of helicity and kinking capacity, and no helices were visualized after 8 passages in broth medium.

## A single MreB is sufficient to confer helicity and to initiate kinks

Proteomics analyses were performed to check the abundance of MreBs in *S. citri*. To assess the extent of variations in cytoskeleton protein abundance in *S. citri*, samples corresponding to different stages of growth, but for which more than 90% of the cells were helical and motile, were compared (Supplementary Table 2). No abundance variation was observed for all MreBs and fibril in motile, helical cells during *S. citri* growth, with the following order of protein abundance: MreB5 > MreB4 > Fib > MreB3 > MreB1 > MreB2.

Because subculturing in SP4 liquid medium led to a decrease in the number of helical, motile cells in transformants, protein abundance was studied after 4 and 6 or 8 passages (4 P, and 6 P or 8 P) in one clone of *Mcap*$^{mreB1-5}$ (cl. 7.5) and two clones of *Mcap*$^{mreB1-5-fib}$ (cl. 8.7 and 32.1) (Table 1). In cultures used for LC–MS/MS analysis, 55% (4 P) and 50% (8 P) of elongated *Mcap*$^{mreB1-5}$ cells were helices, and approximately 8% (4 P) and 6% (8 P) of them showed kinking ability. In *Mcap*$^{mreB1-5-fib}$ clone 8–7, approximately 50% of the cells were helical, of which 18% and 10% were kinking at 4 P and 8 P, respectively. The number of helical, motile cells decreased more rapidly by passaging in the second clone 32-1 of *Mcap*$^{mreB1-5-fib}$: approximately 20% of the helical cells were kinking at 4 P, but less than 8% of the cells were helical, and less than 1% of them were kinking at 6 P. MreB5 was the most highly expressed among MreBs in Mcap$^{mreb1-5-fib}$ and in Mcap$^{mreb1-5}$, as in *S. citri* (Table 1 and Supplementary Table 2). Although all *mreB* genes were added to all clones, not all MreBs were detected by proteomics in *Mcap*$^{mreB1-5-fib}$, even at early passages. Comparison of abundance levels in *Mcap*$^{mreB1-5-fib}$ and in *Mcap*$^{mreB1-5}$ suggested that the production of *fib* could be associated with a decrease in MreB2 and MreB3. Additionally, the set of detected proteins was different from one clone to another: for instance, MreB4 was significantly detected at 4 P in one clone of *Mcap*$^{mreB1-5-fib}$ and not in the other. A similar observation was made for MreB2. Fib was also detected at a significantly lower level in *Mcap*$^{mreB1-5-fib}$ than in *S. citri*, and the MreB total amount significantly decreased in both clones after passaging in SP4. This decrease could be responsible for the loss of helicity and kinking capacity observed after passaging. More generally, the relative abundance of MreBs and Fib in *Mcap*$^{mreB1-5-fib}$ did not mimic those in *S. citri* even at early passages. The absence of MreB3, MreB2, MreB4, or Fib in some clones shown to be able to form kinking helices suggested that these proteins could be dispensable for helicity and for kink generation. It should be noted, however, that the apparently lacking MreBs might be expressed at a level too low to be detected. Additionally, with the use of proteomics herein as a global approach to provide information on the protein content of a whole population, these analyses might not reflect the protein content at the single cell level. Nevertheless, it was clear that MreB5 and MreB1 were most abundant in *Mcap* transformants showing kinking helices. Consequently, to determine the minimal gene set required for helicity and/or motility, a transformant carrying only either *mreB*5 or *mreB*1 was constructed (Supplementary Fig. 6).

While *Mcap* transformation efficiencies with all the constructs (excluding pMT85-PStetM-fib) were rather low (Supplementary Table 3), the transformation efficiency with pMT85-PStetM-*mreB5* was even lower. Nevertheless, two clones could be recovered after three transformation assays. The *Mcap*$^{mreB5}$ population showed a large diversity of shapes from coccoid to filamentous, including 5–10% helices (Fig. 3b and Supplementary Fig. 2) with a helical pitch of 0.97 ± 0.44 μm, which was not significantly different from those of *S. citri* helices (Fig. 3a). Short helices with a length less than 5 μm were also endowed with movements associated with the propagation of membrane deformations along the cell body (Fig. 4). Although the change in helicity could be visualized (Supplementary Movie 3), the helicity was not conserved on the whole length during movements, which led to the transient formation of unwound and untangled

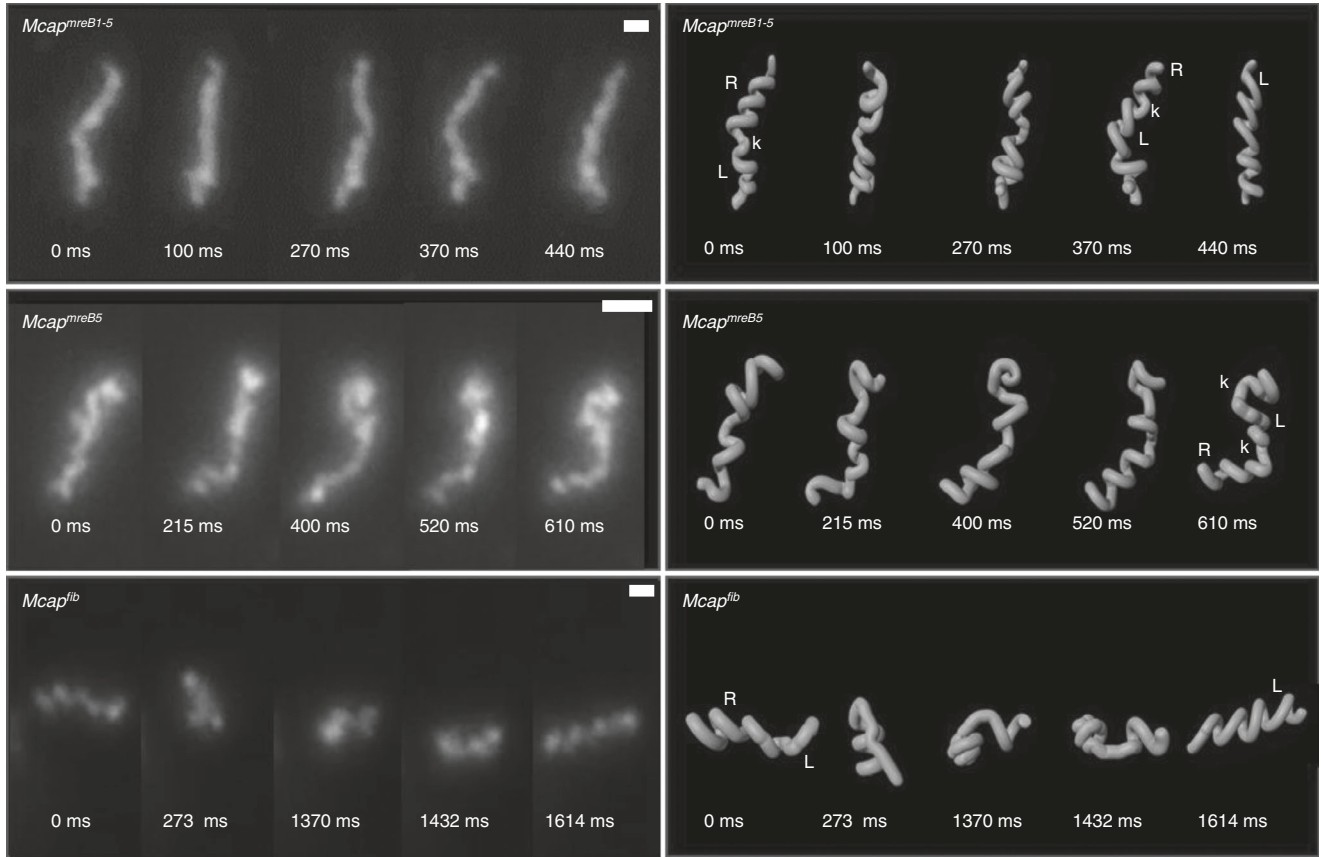

**Fig. 4 | Cell movements in *Mcap* transformants.** Left: Darkfield microscopy time-lapse images showing helicity changes due to the propagation of membrane deformations in *Mcap* transformants bearing all *mreB* genes (top), *mreB5* only (middle) or *fib* only (bottom). The micrographs are representative of at least 3 independent experiments. Right: Proposed simplified models for morphology changes starting from movies from which darkfield images on the left were extracted. Models were built using Blender-3D. Examples of left- (L) and right- (R) handed helices are indicated; k stands for kinks. Note that *Mcap* transformants having *mreB5* only or *fib* only lose their helicity upon propagation of the kink-like membrane deformation. Scale bar: 1 μm.

**Table 1 | Normalized abundance in percentage of MreBs and Fib in *Mcap* transformants**

| | Passage[a] | MreB1 | MreB2 | MreB3 | MreB4 | MreB5 | Fib |
|---|---|---|---|---|---|---|---|
| *Mcap*^control^ | 4 P | ND | ND | ND | ND | ND | ND |
| *Mcap*^mreB1-5^ (cl. 7-5) | 4 P | 0.51 ± 0.07 | 0.19 ± 0.03 | 0.34 ± 0.01 | 0.03 ± 0.01 | 1.47 ± 0.15 | ND |
| | 8 P | 0.47 ± 0.06 | 0.11 ± 0.03 | 0.18 ± 0.003 | 0.03 ± 0.02 | 1.45 ± 0.015 | ND |
| *Mcap*^mreB1-5-fib^ (cl. 8-7) | 4 P | 0.48 ± 0.11 | ND | ND | 0.20 ± 0.02 | 1.97 ± 0.037 | 0.28 ± 0.05 |
| | 4 P | 0.52 ± 0.07 | ND | ND | 0.24 ± 0.07 | 2.26 ± 0.051 | 0.34 ± 0.04 |
| | 8 P | 0.14 ± 0.01 | ND | ND | 0.07 ± 0.004 | 0.56 ± 0.03 | ND |
| *Mcap*^mreB1-5-fib^ (cl. 32-1) | 4 P | 0.80 ± 0.15 | 0.30 ± 0.02 | ND | ND | 2.02 ± 0.51 | 0.36 ± 0.02 |
| | 4 P | 0.68 ± 0.10 | 0.20 ± 0.05 | ND | ND | 1.60 ± 0.25 | 0.30 ± 0.02 |
| | 6 P | 0.11 ± 0.01 | ND | ND | 0.04 ± 0.03 | 0.38 ± 0.04 | ND |

[a] Proteins were detected in bacteria subcultured 4, 6 or 8 (4 P, 6 P, 8 P) times after transformation in axenic medium (4, 6 or 8 passages). For *Mcap*^mreb1-5-fib^ 4 P (clones 8-7 and 32-1), two biological replicates were analyzed. Values are expressed as mean ± SD, with three technical replicates for each clone carrying a combination of cytoskeleton genes. The absence of MreB and Fib in *Mcap*^control^ was checked on two technical replicates. ND: Not detected.

filaments. Longer helices were not found to propagate kinks. Subculturing of *Mcap*^mreB5^ clones led to loss of helicity and motility.

When *mreB5* was coexpressed with *fib* (*Mcap*^mreB5-fib^) (Supplementary Fig. 3), long helices were produced, and the helical pitch was 0.77 ± 0.13 μm (Fig. 3). However, kink propagation was not observed with the gene combination *mreB5-fib*.

Altogether, these observations showed that the addition of *mreB5* was sufficient to confer helicity to *Mcap* and triggered kink-like membrane deformations for short helices. The morphology of *Mcap*^mreB5^ was then compared to those of *Mcap*^mreB1^ carrying the *mreB1* gene alone (Fig. 3 and Supplementary Fig. 2). The latter showed helices with a larger pitch (1.81 ± 1.08 μm) (Fig. 3a). Kink-based membrane movements of helices were not recorded with *Mcap*^mreB1^. The addition of the *fib* gene together with *mre*B1 (Supplementary Fig. 7) produced long, nonkinking helices with a mean pitch of 1.33 ± 0.98 μm (Fig. 3 and Supplementary Fig. 2).

## Expression of MreBs and Fib is associated with the formation of cytoskeletal structures that interact with the plasma membrane

CryoEM was used to assess whether the different gene combinations led to the formation of stable cytoskeletal filaments in *Mcap*. Figure 5a illustrates the tapered end of *S. citri* GII-3 and its internal cytoskeleton closely associated with the membrane at locations with negative curvature and showing a regular striated pattern with a $5.2 \pm 0.7$ nm periodicity (Supplementary Fig. 8). An image of a typical control cell corresponding to *Mcap* transformed with the empty vector lacking any internal cytoskeletal structure is shown in Fig. 5b. Upon adsorption on carbon grids and cryofixation prior to cell imaging, most helical *Mcap* recombinants lost their morphology. Nonetheless, the cells showed internal cytoskeleton elements. No cell polarization could be observed in *Mcap* recombinants regardless of the gene combination.

As explained above, addition of the complete gene set (*mreB1-5* together with *fib*) in *Mcap* resulted in different protein abundance profiles. Here, clone 8.7 (4 P) expressing MreB1, MreB4, MreB5, and

Fib was analyzed using cryoEM. Expression of these proteins induced the production of internal fibers positioned next to the membrane (Fig. 5c). The cytoskeleton could recover the whole inner side of the membrane or appear only next to negatively curved membrane areas. The latter organization mimicked that of *Spiroplasma*. The internal cytoskeletal structure in *Mcap*$^{mreB1-5-fib}$ corresponded to a stack of filaments with a mean width ranging from 4 to 10 nm and a regular striated pattern with a $6.3 \pm 0.9$ nm periodicity (Supplementary Fig. 8b).

An internal structure was also observed in *Mcap*$^{mreB1-5}$, *i.e.*, in the absence of *fib* (Fig. 5d). The internal filament formed by one or several MreBs ran next to the membrane, in close association either with curved membrane areas or with a larger membrane zone. The width of the protofilaments ranged from 4 to 8 nm, with a $5.2 \pm 0.4$ nm periodicity (Supplementary Fig. 8c).

*Mcap* recombinants expressing a single cytoskeleton protein (MreB1 or MreB5) also showed internal structures. Expression of MreB1

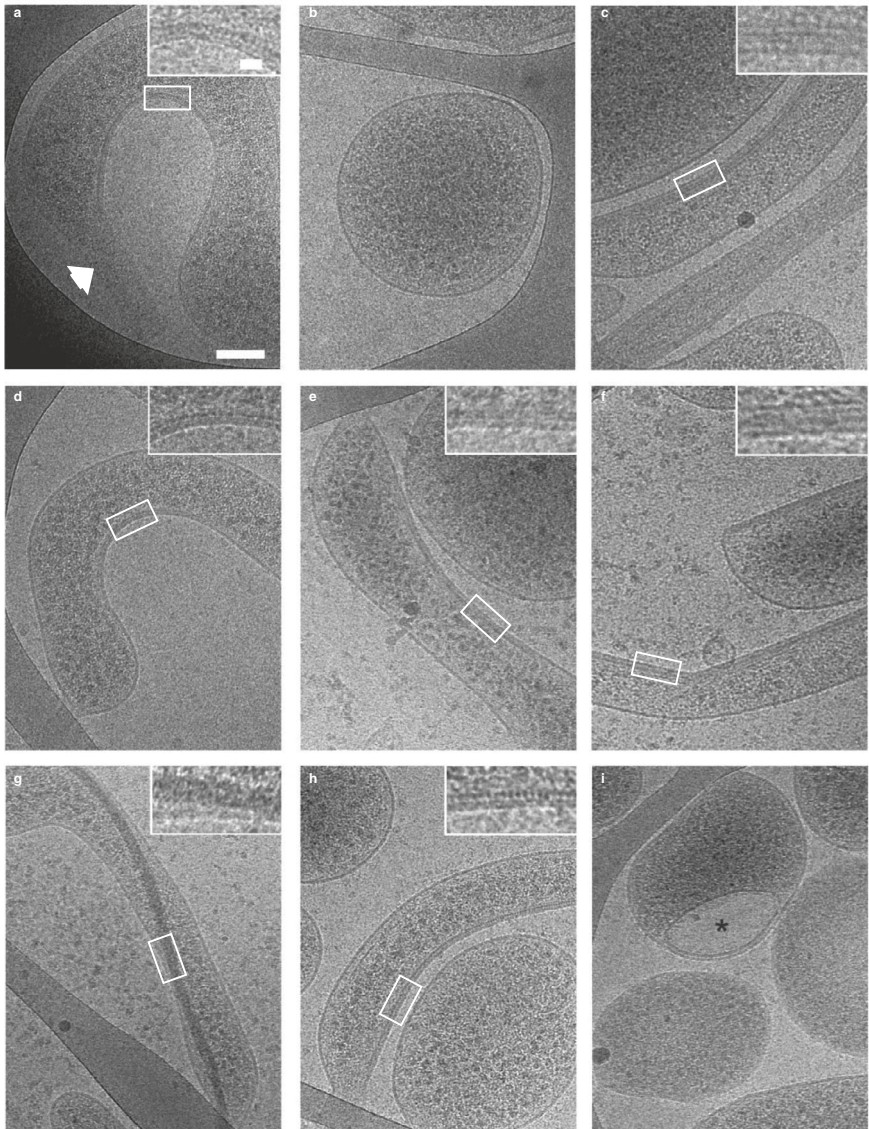

**Fig. 5 | 2D cryo-electron microscopy reveals the formation of cytoskeletal filaments in *Mcap* transformants bearing *S. citri* cytoskeleton genes.**
**a** *Spiroplasma citri* cytoskeletal fibers are localized in the cytoplasm next to negatively curved areas of the plasma membrane. Note the tapered (duckbill-shaped) tip of the cell indicated by the white arrow. **b** *Mcap*$^{control}$ cells transformed with the empty vector do not show any cytoplasmic cytoskeleton fibers. **c**–**h** Cytoskeletal

fibers were imaged in *Mcap*$^{mreB1-5-fib}$ **c**, *Mcap*$^{mreB1-5}$ **d**, *Mcap*$^{mreB1}$ **e**, *Mcap*$^{mreB1-fib}$ **f**, *Mcap*$^{mreB5}$ **g**, and *Mcap*$^{mreB5-fib}$ **h** cells. **i** Internalized vesicle (indicated by *) observed in some *Mcap*$^{mreB5}$ cells. Inset, magnified view of the region outlined by the white box in each image. Scale bars: 100 nm and inset 20 nm. These images are representative of at least 3 independent experiments, except with *Mcap*$^{mreB5}$, for which a clear cytoskeleton was visible in a single cell (see main text).

induced the formation of 4-8 nm cytoskeletal fibers adjacent to the membrane (Fig. 5e). These internal structures could recover the entire inner side of the membrane. In *Mcap^mreB5*, the MreB5 filament was clearly observed only in a single cell (Fig. 5g). The MreB5 filament, having a width of up to 27 nm, crossed the whole cell body with localized interactions with the plasma membrane. When MreB5 was coexpressed with Fib, internal filaments showed the striated pattern (periodicity of 5.1 ± 0.6 nm, Fig. 5h and Supplementary Fig. 8d) already observed in *S. citri* and in *Mcap^mreb1-5*. Interestingly, internalized membrane-bound structures were observed in *Mcap^mreB1-5*, *Mcap^mreB1* and *Mcap^mreB5* (Fig. 5i). This observation was consistent with a strong ability of MreB1 and MreB5 to induce membrane curvature at directed membrane areas, as an excessive curvature could lead to inverse membrane blebbing followed by detachment and internalization of a more or less spherical membrane-bound vesicle.

## Discussion

### Identification of the minimal requirements for conferring helicity and motility to *Mcap*

The *Spiroplasma* cytoskeleton was reconstituted in a mycoplasma that was initially pleomorphic and nonmotile. Heterologous expression of MreBs and fibril proteins in *Mcap* cells changed the cell morphology from spherical to elongated cell bodies. In addition, it was sufficient to confer helicity and kinking movements to the mycoplasma cell. All *Spiroplasma* species have at least 5 copies of the *mreB* gene[22], which raises the question of the redundancy of functions between the different isoforms. Fib, MreB5 or MreB1 could generate cellular helicity in *Mcap*, indicating that each of these proteins was able to induce the membrane curvature required for helicity in *Mcap*. The combination of Fib with MreB1 allowed the tightening of the helices, but the mean helical pitch remained higher than those of *S. citri* cells. In contrast, MreB5 expression produced helices with a mean pitch similar to those found in *S. citri*. Hence, each of these proteins (Fib, MreB1 or MreB5) represented a minimal requirement for helicity in a wall-less bacterium, and MreB5 represented the minimal requirement to mimic *Spiroplasma* helicity.

Regarding motility, the expression of MreB1 did not induce a membrane deformation that propagated along the cell body, while MreB5 was able to confer kink-based movements to short helices. Therefore, a single MreB, such as MreB5, represented the minimal requirement to produce helical cells endowed with kinking ability. However, the cytoskeleton resulting from the expression of a single MreB was not sufficient to allow conservation of the cell length and helicity upon propagation of membrane deformation (Fig. 3). Changes in cell length were likely responsible for the loss of helix directionality. A similar phenotype was obtained when transforming *Mcap* with *fib* only. In *Spiroplasma*, fibril forms polymers with a consistent length during motility[21], ensuring efficient conservation of the helical shape when the kink travels along the cell. In *Mcap*, sole addition of *fib* did not provide the required capacity to preserve a constant length during propagation of the kink. As discussed above, coexpression of Fib with MreB5 allowed the production of long helices, suggesting that MreB could help stabilize the Fib-based cytoskeleton. However, in this configuration in *Mcap* (MreB5 + Fib), the energy transfer to the membrane coupled with MreB5 and Fib was not sufficient to induce membrane deformation propagation along the entire cell length, which could be due to the limited efficacy of the Fib-MreB5 combination in cell deformation. This combination may have produced more rigid cells in response to the presence of a Fib filament compared with recombinants expressing a single *Spiroplasma* cytoskeletal protein. This hypothesis must be assessed by measuring stiffness in different transformants in future studies. While MreB4 was the second most abundant MreB isoform in *S. citri*, it was absent in kinking *Mcap^mreB1-5-fib* populations. In contrast, MreB1 was highly expressed. This observation provides some clues in favor of the hypothesis of at least partially overlapping functions of MreB1 and MreB4 in *Spiroplasma*. The phylogenetic tree based on MreB sequences from 26 *Spiroplasma* species revealed that these proteins could be classified into 5 clusters[23]. However, several MreBs, including *S. citri* MreB1, could not be clearly classified as MreB1 or MreB4, arguing in favor of our hypothesis of overlapping functions between these isoforms. The expression of MreB5 in all helical and motile *Mcap* transformants was in agreement with a major role played by this specific MreB in *Spiroplasma* helicity and motility. This result is consistent with the restoration of helicity and motility observed upon complementation with the *mreB5* WT gene in *S. citri* ASP-1[23]. Notably, the absence of a functional *mreB5* gene in the ASP-1 strain could not be compensated by any of the other 4 *mreB* genes, strongly suggesting the presence of a functional specialization between MreB5 and the other isoforms.

### Insights into the structure of *Spiroplasma* MreB filaments

*Spiroplasma* MreB3 and MreB5 have been previously shown to polymerize in vitro[23,25,26], and Masson et al.[24] recently demonstrated that *Spiroplasma* MreBs expressed in *E. coli* can form a complex network, supporting the hypothesis that the different isoforms participate in the production of a cytoskeleton in *Spiroplasma*. The present results not only indicate that *Spiroplasma* MreBs polymerize into stable filaments in a wall-less bacterium that is phylogenetically close to *Spiroplasma*, but they also provide clues regarding the cytoskeletal organization. In *Mcap*, the width of MreB filaments could be as thin as 4 nm, similar to that of unidentified filaments previously observed in *S. citri*[32], which validates our heterologous system. In rod-shaped, walled bacteria, MreBs establish a direct interaction with the plasma membrane[33]. The capacity of MreB5 to interact with lipid bilayers in vitro has been previously demonstrated[23]. In *Mcap*, both MreB1 and MreB5 induced a plasma membrane curvature and formed filaments that were closely associated with the membrane. This ability to distort membranes is common to some other MreBs, as shown for TmMreB, an MreB from the thermophilic bacterium *Thermotoga maritima*[33]. In some cells, the expression of each MreB in *Mcap* led to the internalization of membrane-bound vesicles. Interestingly, similar vesicles were observed in *E. coli* expressing TmMreB[33], strengthening the hypothesis that *Spiroplasma* MreB1, MreB5 and TmMreB share functional features. In the absence of any resistance provided by a peptidoglycan wall in Mollicutes, the curvature induced by *Spiroplasma* MreBs could lead to the formation of a helical cell. Although both MreB1 and MreB5 seemed to interact directly with the membrane, analyses of helical pitches in the different constructs led us to propose the hypothesis that MreB5 is the main determinant of the cell curvature allowing the correct localization of interactions between the internal ribbon and the plasma membrane. To induce the formation of helices with a regular pitch, *Spiroplasma* MreBs must interact with specific membrane partners. Considering that MreB5 interacts with liposomes[23], its membrane partner in *Spiroplasma* and in *Mcap* may be of a lipidic nature. Since anionic phospholipids exclude assembled MreB in *E. coli*[34], MreB1 and B5 could preferentially interact with anionic phospholipid-depleted membrane areas in *Mcap* and *Spiroplasma*. A heterogeneous distribution of phosphatidylglycerol and cardiolipin, major anionic phospholipids in the *Mcap* and *Spiroplasma* membranes[30], could occur and favor the interaction of *Spiroplasma* MreBs with membrane parts enriched in specific lipids. Notably, bundles of thick filaments were associated with a straight cell morphology, indicating the likely requirement of a thinner filament for helicity. Additionally, our cryoEM experiments indicated that MreB1 and B5 filaments could span the whole *Mcap* cell length over several micrometers. In most rod-shaped bacteria, the current prevalent hypothesis assumes that MreBs form short membrane-associated filaments[5], but their length is still debated[35]. The lengths of MreB1 and B5 filaments in *Mcap* may differ from those of MreB filaments in *Spiroplasma*, as their expression levels are probably not the same in the two species.

## Role of MreBs in motility

Demonstration of a role of MreB in bacterial motility has thus far only been obtained in the gliding motility of *Myxococcus xanthus*[36]. A role for MreB5 in *Spiroplasma* swimming has also been suggested since complementation of the nonhelical, nonmotile *S. citri* ASP-1 with *mreB5* from the helical and motile *S. citri* GII-3 restores not only helicity but also kinking motility[23]. Nonetheless, *Spiroplasma* kinking motility requires helicity. Therefore, the motility restoration in *mreB5*-complemented ASP-1 could be due to helicity recovery, and the role played by MreB5 in motility could subsequently be indirect. Differential polymerization kinetic parameters of MreB isoforms measured in vitro led Sasajima and collaborators[21] to propose a theoretical model for the *Spiroplasma* swimming mechanism in which two MreBs would generate a force similar to that of a bimetallic strip. This force would then be transmitted to fibril polymers and result in a change in handedness of the helical fibril filaments. However, a single MreB, MreB5, was also able to endow *Mcap* cells with movements resulting from the spread of membrane deformation along the cell body. In most species, MreB filaments are highly dynamic polymers that align along the greatest principal membrane curvature[37] and show circumferential motion following a path that is mostly perpendicular to the long axis of the cell[38]. In rod-shaped, walled bacteria, this motion is powered by the peptidoglycan synthesis machinery[39]. Without peptidoglycan synthesis enzymatic activity in Mollicutes, assembly and dynamics of MreBs may drive the initiation and propagation of membrane deformations at the cell surface. The presence of 5 to 8 MreB isoforms in *Spiroplasma* (compared with 1 to 3 MreB homologs in model rod-shaped, walled bacteria such as *E. coli* or *B. subtilis*) could allow the generation of a cumulative force that is transmitted to the membrane.

## Role of fibril in *Spiroplasma* shape and motility

The present work also sheds light on the possible involvement of the fibril in the generation of helicity, its maintenance and motility. Fib filaments were observed in close interaction with the *S. citri* plasma membrane in previous studies[18,40]. It was long thought that fibril was responsible for *Spiroplasma* motility by changing its length[40]. However, recent studies indicate that the length of Fib polymers does not change during cell movement[41]. The structure of Fib filaments has been studied by electronic microscopy: Fib filaments show a helicity with a pitch close to those of the *Spiroplasma* cell, leading to the conclusion that Fib is the determinant of helicity in *Spiroplasma*[41]. Our results indicated that Fib induced membrane curvature in the absence of any MreB. However, the helix constructed with Fib only was in a more relaxed form than those observed when MreB5 and Fib were coexpressed, suggesting that MreB5 could help position the Fib filament by interacting with both Fib and the membrane. This finding is in line with the ability of MreB5 to bind both fibril and liposomes in vitro[23]. Given its well-adapted helical structure, Fib could be a major determinant of helical shape maintenance. Regarding long helix generation, Fib had to be associated with MreB5 in *Mcap* and probably in *Spiroplasma*. Our work also provides some clues about the role played by Fib in motility. Fib allowed *Mcap* to form kinking helices very similar to those observed with MreB5. This result is intriguing because Fib and MreB do not share any sequence similarity, and unlike MreBs, Fib lacks ATPase activity[41]. In the most recently proposed model of *Spiroplasma* motility, MreBs transmit force to the fibril filament, which changes its handedness to generate a shift in the cell body helicity[21]. We propose that fibril is not only an essential structural component for the transmission of torque to the membrane but also participates in the generation of the required force. Interestingly, the genome of a few motile *Spiroplasma* species, including *S. sabaudiense*[42], lacks the *fibril* gene[43] but contains more than 5 *mreB* genes[12,22], including several copies of MreB5[22]. It is therefore tempting to suggest an evolutionary convergence between MreB5 and fibril to ensure swimming motility in spiroplasmas.

## Lack of efficient swimming in *Mcap* recombinants

Notably, none of the *Mcap* recombinants exhibited translational motility in liquid medium. The most plausible explanation is that, in *Mcap*, the relative abundance of proteins did not match those observed in *S. citri*. More specifically, Fib was detected at a lower level in *Mcap*, and MreB3 and MreB4 were not detected. A proper stoichiometry of these proteins is likely to be required for optimal cell stiffness, stability of the helical structure in *Mcap* recombinants during propagation of the kink, and efficient swimming. Thus, the lack of unidentified regulators maintaining the required stoichiometry among cytoskeletal proteins in *Mcap* may be responsible for the lack of translational motility in the transformants. Although this hypothesis is preferred, it is based on global proteomics performed on a whole population, which does not necessarily reflect the relative expression in single cells. Indeed, the ability of Mcap*mreB1-5,fib* to produce a large range of morphologies, including straight and flexible ones, could be due to differences in the amounts of cytoskeletal proteins at the single-cell level. In addition, kink propagation was observed in approximately 10% of Mcap*mreB1-5,fib* elongated cells, which led us to propose a second hypothesis: propagation of the kinks was not sufficient to propel the cells in liquids. In *Spiroplasma*, swimming velocity has been proposed to depend, for a fixed pitch angle, on the ratio of the distance between kinks to the cell length[44]. It is possible that the requirements in terms of kink frequency are not met in *Mcap* transformants. Among other hypotheses that could explain the lack of motility in liquids is the lack of cell polarization in recombinants. Indeed, the tapered shape of the *Spiroplasma* tip may favor penetration and propulsion in liquid media. Finally, it is reasonable to assume that firm attachment of the internal cytoskeleton ribbon at the two poles of the cells is required to ensure translational movement. In *Mcap* recombinants, such attachment may be lacking. In addition, the *Spiroplasma* dumbbell-shaped core[19] was absent in *Mcap* transformants. We could show here that this structure was not required for kink generation and propagation, but it may be essential to generate an initial kink properly located at the cell extremity.

## Proposed model for helicity and motility mechanisms in *Spiroplasma*

The present study demonstrates the importance of the structure and organization of MreBs and fibril in determining helicity and motility in *Spiroplasma*. These findings provide some clues that lead to a structural model in which (i) MreB5 filaments interact specifically with membrane areas enriched in unidentified lipids and induce membrane curvature; (ii) MreB5 determines the correct localization of MreB1 and fibrils, which both participate in membrane curvature; and (iii) MreB1, MreB5 and Fib polymerize into filaments along the plasma membrane following the shortest path of the cell body, and the MreB/Fib association stabilizes the copolymer structure. The fibril transmits the cumulative force generated by MreBs and itself to the membrane to form and propagate the kink along the cell helix. The role of the other MreBs (B2, B3, and B4) remains to be elucidated. As discussed above, it is plausible that MreB4 also constitutes the cytoskeletal ribbon and that the functions of MreB1 and MreB4 overlap. This study opens up new perspectives, particularly for understanding the swimming mechanism in *Spiroplasma*. Now that the involvement of MreBs in motility has been demonstrated, it appears essential to understand how the expression of MreBs is regulated during the cell cycle to distinguish their functions associated with division from those linked to the maintenance of helicity and motility. Further studies will also be required to elucidate the mechanism of force generation by the cytoskeleton components and its transmission to the membrane, as well as to determine whether the tapered shape of the *Spiroplasma* end allows propulsion of the cell in liquid medium.

# Methods

## Bacterial strains and culture conditions

*Escherichia coli* (NEB 10-beta Electrocompetent *E. coli* or NEB 5-alpha Competent *E. coli* High Efficiency) served as host strains for the cloning experiments and plasmid propagation. Plasmid-transformed *E. coli* cells were grown at 37 °C in Lysogeny Broth (LB) or on LB agar supplemented with ampicillin at 100 µg/mL and tetracycline at 5 µg/mL.

The restriction-free *M. capricolum* subsp. *capricolum* strain California KidT (*Mcap*) was used in this study. This strain was obtained by inactivation of the CCATC restriction enzyme in the wild-type strain (ATCC 27343)[28]. *Mcap* and its derivatives were grown at 37 °C in SP5 medium[27,45] supplemented with 5 µg/mL tetracycline. One passage corresponded to the transfer of 10 µL of mycoplasma culture (at pH-6.5) to 1 mL of mycoplasma medium (1/100 dilution) followed by an incubation period of ≥24 h (depending on the constructions) at 37 °C. SP5 medium containing 0.8% noble agar was used to grow colonies of *Mcap* and its derivatives.

*S. citri* strain GII-3 was originally isolated from its leafhopper vector *C. haematoceps* captured in Morocco[46]. Spiroplasmas were cultivated at 32 °C in SP4 medium. SP4 containing 0.8% noble agar was used to grow colonies of *S. citri*.

## Live/dead viability assays

We performed nucleic acid staining using a LIVE/DEAD™ BacLight™ Bacterial Viability Kit L7012 (Invitrogen™ by Thermo Fisher Scientific, Life Technologies Corporation 29851 Willow Creek Road, Eugene, Oregon 97402) containing the fluorochromes propidium iodide (PI) and Syto™9.

A total of 250 µL of culture was mixed with 0.2 µL of Syto™9 and 0.2 µL of PI. The cells were then incubated in the dark for 15 minutes. Five microlitres of the stained bacterial suspension was pipetted onto a glass slide and covered with a coverslip. Glass slides and coverslips were sealed using a coverslip sealant (CoverGrip™, Biotium).

Microscopic observations were performed with a ZEISS Axio Imager M2 with a 63× oil immersion objective of 1.4 numerical aperture. Images were acquired using Micro-Manager open source microscopy software[47] and a CoolSNAP™ HQ2 Monochrome camera from Photometrics™. Syto™9 and PI were observed with the following two excitation and emission filters: 442–502 nm/485–555 nm and 522–602 nm/longpass 593 nm. The exposure time was adjusted to avoid saturation, and the brightness and contrast were adjusted using Fiji[48] (version 2.35).

## Plasmid construction

Seven plasmids were built during this study. All were derived from the transposon-based plasmid pMT85-PStetM (4.73 kbp)[49–51], which harbors the *tet*(M) gene from transposon Tn916 under the control of the spiralin promoter (PS).

Five plasmids were constructed using the NEBuilder® HiFi DNA Assembly Cloning Kit (Supplementary Table 4 and Supplementary Figs. 1, 5–7). Depending on the assemblies, two to four overlapping DNA fragments were PCR amplified (Advantage HF 2 PCR Kit from Clontech) using the primers described in Supplementary Data 1, purified and combined at 50 °C according to the manufacturer's instructions. DNA cassettes were designed to contain ~40 bp overlaps. One DNA cassette corresponded to the whole transposon-based plasmid pMT85-PStetM. Others corresponded to different regions of the *S. citri* GII-3 genome: the *fibril* gene (SPICI12_006), the *mreB1* gene (SPICI13_009) and the locus composed of *mreB2*, *mreB3*, a hypothetical protein encoding gene (HP), *mreB4* and *mreB5* (SPICI01A_045 to SPI-CI01A_049). *S. citri* DNA cassettes were amplified to conserve the native promoters.

The two last plasmids (pMT85PStetM-mreB1 and pMT85PStetM-*mreB5*), which were derived from plasmids pMT85PStetM-*mreB1-fib*

and pMT85PStetM-*mreB5-fib*, respectively, were built using the Q5® Site-Directed Mutagenesis Kit Protocol (Supplementary Table 4).

All oligonucleotides for plasmid construction were supplied by Eurogentec and are described in Supplementary Data 1. Prior to being used for transformation into *Mcap*, the purified plasmids were verified by restriction analyses and sequencing.

## Mycoplasma transformation and screening

*Mcap* was grown in SOB + medium at 30 °C and transformed using the established 5% polyethylene glycol (PEG)-mediated protocol[27]. Transformations were conducted using 5 to 10 µg of plasmids, and transformants were grown on selective medium SP5 plus tetracycline 5 µg/mL (SP5 tet5) for 7 to 21 days depending on the construction. They were then propagated in SP5 plus tetracycline 5 µg/mL.

Colonies obtained on selective plates were picked and transferred into 1 mL of SP5 tet5 liquid medium and incubated at 37 °C. After three passages, 200 µL was used for DNA extraction and PCR analysis, and 800 µL was stored at −80 °C.

Transformant genomic DNA was extracted with the NucleoSpin® Tissue kit (Macherey-Nagel, Düren, Germany) and further analyzed by (i) PCR to verify the presence of the *tet*(M) and *S. citri* genes (*fibril*, *mreb1*, *mreb2-5*) and (ii) direct sequencing to localize the transposon insertion site (see below).

The primers used for PCR, direct sequencing and localization of insertions into *Mcap* genomes are summarized in Supplementary Data 1.

## Determination of the transposon insertion site by single-primer PCR

Transposon insertion sites were determined by single-primer PCR. The 25-µL final reaction volume contained 1× PCR Buffer (NEB), 3 mM MgCl₂, 1 µM SPP2-pMT85-TetM primer, 0.2 mM dNTPs, 0.5 U of Taq NEB polymerase (NEB), and 2.5 µL of the transformant DNA. The PCR amplification cycle was performed as previously described[52].

Transposon insertion sites were determined by Sanger sequencing of the PCR products with the nested primer MT85-1 (Supplementary Data 1).

## Dark-field microscopy and cell length measurements

One volume of cultures of exponentially growing *Mcap* in SP4 tet5 and of *S. citri* in SP4 (pH 6.8–6.9) was diluted in one volume of fresh medium. Bacterial suspensions were prepared between two microscope slides sealed using nail polish, with a liquid thickness of 15 µm. The morphology of *Mcap* transformants growing in SP4 media was observed using an Eclipse Ni (Nikon) microscope working in reflection and equipped with a dark field condenser. The Nikon oil immersion microscope objective was 60× with a numerical aperture (N. A.) of 0.80. Images were acquired with an Iris 9™ Scientific CMOS camera (2960 × 2960 pixels). Videos were recorded at the maximal frame rate of 10 to 30 frames per second (fps) using the software NIS-Elements Br (Nikon). The *Mycoplasma* cell length and helicity parameters were measured from isolated frames using the same software. Cells were considered helical if at least 70% of the total cell length showed helicity; otherwise, they were considered (nonhelical) filamentous cells. Cells were considered branched if at least one branch could be visualized along the main (longest) filament. Helical pitch measurements excluded nonhelical cells. Data are expressed as the mean ± standard deviation (SD). Graphs and statistical analyses were performed using Excel 2019. Statistical significance was estimated using the two-tailed unpaired *t* test. For each construct, data from 3 independent cultures were collected and mixed, and differences between transformants were considered significant at $p < 0.001$. The total number of cells used to draw the graphs is indicated in the corresponding figure captions, and raw data are given in Source data file.

Time-lapse images were isolated from the video recordings. Simplified models shown in Fig. 3 were built using Blender 3D v 3.2.2.

## Cryogenic electron microscopy

Lacey carbon formvar 300 mesh copper grids were used, which first submitted to a standard glow discharge procedure (3 mbar, 3 mA for 40 s). Plunge freezing was achieved using the EM-GP apparatus (Leica). Four microlitres of sample was deposited on the grid and immediately blotted for 2 s with Whatman paper grade 5 before plunging into a liquid ethane bath cooled with liquid nitrogen (−184 °C). The settings of the chamber were fixed at 80% humidity and 20 °C. Specimens were used undiluted in culture medium. They were observed at −178 °C using a cryo holder (626, Gatan) with a ThermoFisher FEI Tecnai F20 electron microscope operating at 200 kV under low-dose conditions. Images were acquired with an Eagle 4k × 4k camera (ThermoFisher FEI) and processed in ImageJ 1.4.3.68.

## Proteomics

*Mycoplasma* and *Spiroplasma* cells were harvested by centrifugation at 10,000 × *g* for 20 min before being washed twice with Dulbecco's phosphate-buffered saline (Eurobio) (mycoplasmas) or with a solution containing 8 mM HEPES and 280 mM sucrose (spiroplasmas). The protein concentration was determined using the DC Protein Assay (Bio-Rad). Fifteen micrograms of protein were mixed with SDS loading buffer (75 mM Tris-HCl, pH 6.8, 50 mM DTT, 2% SDS, 0.02% bromophenol blue, 8.5% glycerol) and then loaded onto an SDS–PAGE stacking gel (7%). Short electrophoresis was performed (10 mA, 45 min and 20 mA, 2 h) to concentrate the proteins. After migration, the gels were stained with Coomassie Blue and destained (50% ethanol, 10% acetic acid, 40% deionized water). The revealed protein band from each fraction was excised, washed with water, and then immersed in a reductive solution (5 mM DTT). Cysteines were irreversibly alkylated with 20 mM iodoacetamide in the dark. Following the washing and drying steps, the gel bands were subjected to protein digestion with trypsin added to a final protease to protein ratio of 1:25 for 3 h at 37 °C in ammonium bicarbonate buffer (10 mM and pH 8). Peptides were extracted with 50% CH$_3$CN, followed by 0.1% TFA and 100% CH$_3$CN. The collected samples were then dried. Peptides were then analyzed by mass spectrometry. All experiments were performed on an LTQ-Orbitrap Elite (Thermo Scientific) coupled to an Easy-nLC II system (Thermo Scientific). One microlitre of sample (200 ng) was injected onto an enrichment column (C18 Acclaim PepMap100, Thermo Scientific). The separation was performed with an analytical column needle (NTCC-360/internal diameter: 100 µm; particle size: 5 µm; length: 153 mm, NikkyoTechnos, Tokyo, Japan). The mobile phase consisted of H$_2$O/0.1% formic acid (FA) (buffer A) and CH$_3$CN/FA 0.1% (buffer B). Tryptic peptides were eluted at a flow rate of 300 nL/min using a three-step linear gradient: from 2 to 40% B over 76 min, from 40 to 100% B over 4 min and at least 10 min at 100% B. The mass spectrometer was operated in positive ionization mode with the capillary voltage and source temperature set at 1.8 kV and 275 °C, respectively. The samples were analyzed using the CID (collision induced dissociation) method. The first scan (MS spectra) was recorded in the Orbitrap analyser (*r* = 60,000) with the mass range *m/z* 400–1800. Then, the 20 most intense ions were selected for tandem mass spectrometry (MS$^2$) experiments. Singly charged species were excluded from the MS$^2$ experiments. Dynamic exclusion of already fragmented precursor ions was applied for 30 s, with a repeat count of 2, a repeat duration of 30 s and an exclusion mass width of ±5 ppm. Fragmentation occurred in the linear ion trap analyser with a collision energy of 35. All measurements in the Orbitrap analyser were performed with on-the-fly internal recalibration (lock mass) at *m/z* 445.12002 (polydimethylcyclosiloxane). The resulting raw files were analyzed with Proteome Discoverer 1.4.1.14 (Thermo Scientific). A database search was performed with the Mascot (version 2.8.0)

algorithm against "GCF_000012765.1_ASM1276v1_protein" (801 entries) retrieved from NCBI and containing the protein sequences encoded in the *M. capricolum* subsp. *capricolum* genome and a multifasta containing *S. citri* GII-3 and plasmid-encoded protein sequences. The following search parameters were used: trypsin was specified as the enzyme allowing for one miscleavage; carbamidomethyl (C) and oxidation (M) were specified as variable modifications; the precursor mass range was set between 350 and 5000 Da, with a precursor mass tolerance and a fragment ion tolerance of 10 ppm and 0.35 Da, respectively. Peptide validation was performed using the Percolator algorithm[53], and only "high confidence" peptides were retained, corresponding to a 1% false discovery rate. A minimum of 2 peptide spectrum matches (PSMs) and 2 unique peptides were required to consider protein identification and quantification. For protein abundance, the PSMs of the unique peptides for each protein were summed. Normalization was performed by dividing the total PSMs of each protein by the sum of the PSMs of all identified proteins in a sample. Results are given as a percentage in Table 1 and Supplementary Table 2.

## Reporting summary

Further information on research design is available in the Nature Portfolio Reporting Summary linked to this article.

## Data availability

The mass spectrometry data generated in this study have been deposited to the ProteomeXchange Consortium via the PRIDE partner repository[54] under accession code PXD036290 (http://www.ebi.ac.uk/pride). All *Mycoplasma capricolum* subsp. *capricolum* protein sequences can be found at https://www.ncbi.nlm.nih.gov/search/all/?term=GCF_000012765.1_ASM1276v1_protein. All *Spiroplasma citri* GII3 protein sequences were previously reported[55]. All other data supporting the findings of this study are available within the paper and its supplementary information files. Source data are provided with this paper.

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

## Acknowledgements

We thank Nicolas Martin, Jean-Christophe Baret (Univ. Bordeaux, CNRS-UMR5031, Bordeaux, France), the 'Frontiers of Life' network (Univ. Bordeaux, France) and the Microbiology Community of Nouvelle-Aquitaine (Microbio-NA) for helpful discussions. We also thank Eva Manduchet,

Sebastian Lillo and Bruce Richard for technical help. The BL doctoral fellowship was granted by Univ. Bordeaux through the 'Interdisciplinary projects' call. This research was partially supported by the Agence Nationale de la Recherche (ANR) through grant ANR-22-CE44-0015-01.

## Author contributions

L.B. defined the project; L.B. designed microscopy and biochemistry experiments, and C.L. molecular biology experiments; C.L. and F.R. produced the transformants, and C.L. performed the transformation efficiency measurements and identified the transposon insertion sites; L.B., B.L., Y.D., J.P.D. collected and analyzed darkfield and fluorescence microscopy data; M.D., O.L. and L.B. collected and analyzed the cryoelectron microscopy data; M.H., J.H. and L.B. produced and analyzed mass spectrometry data; L.B., C.L., A.B. drafted the manuscript. All authors reviewed and edited the final manuscript. L.B., C.L., J.P.D. and A.B. contributed to funding acquisition.

## Competing interests

The authors declare no competing interests.
