## [Peer Review File · Nature Communications]

Cytoskeletal components can turn wall-less spherical bacteria into kinking helicesReviewer #1 (Remarks to the Author):

The manuscript is based on a fascinating idea - using gene expression in an unrelated but highly plastic model organism, *Mycoplasma capricolum* to identify a minimal set of cytoskeletal or cytoskeleton-associated components that confers generation of *Spiroplasma citri*'s characteristic helical shape and unusual motility, providing a system to probe the contribution of each component to the spiroplasma phenotype. There are some considerable successes in this work. Using a set of five MreB genes and the gene for the fibril protein, the researchers most assuredly achieved the spiroplasma's shape, as well as the characteristic propagation of the helix kink along the cell's long axis. They also had some partial success at probing the roles of each of the MreB isoforms. These cells did not achieve swimming through liquid, leading the authors to propose that other proteins are required for that process. The methodology is sufficiently well-described to allow reproducibility, and with perhaps one exception (see point #1) is appropriate. There are some elements of the manuscript that are confusing, and some of the authors' data lead me to believe that alternative interpretations of their results might better explain the phenomena the researchers observe.

The biggest problems:

1) LC-MS data suggest that in different transformants, the amounts of each MreB isoform differ, and in most instances some isoforms are undetectable. Unfortunately, the absence of a transformant in which all isoforms plus fibril are produced, together with the fact that those that are produced are in wildly varying ratios, makes it quite difficult to establish a clear relationship between the presence of these proteins and the observed phenotypes. All things considered, rather than invoking additional factors required for swimming, it seems to me that the most straightforward explanation for the absence of swimming is that the MreB isoforms aren't all there in their normal ratios, so the structure that's assembled isn't normal. According to this, what's really missing is factors that generate and maintain the normal stoichiometry of these proteins in *S. citri*. Has that been characterized?

2) I'm confused by the use of the word "motility." Sometimes it seems to refer to swimming motion through the media. Sometimes it seems to refer to kink propagation. Sometimes I'm not really sure what it means. In this case, I would call movement of the whole cell "motility," and propagation of the kink "kink propagation."

3) Relatedly, there is a significant logical flaw in the way the authors describe the relationship between kink propagation and swimming motility. Surely, in prior literature it has been hypothesized that kink "propagation is responsible for the movement of the bacteria," as the authors state on line 75 and reiterate on line 200. However, the authors' own results clearly dismiss that model, as they have achieved kink propagation but not motility. This is an important point that they have not emphasized; it isn't actually clear that it's been considered.

4) I don't think it's appropriate to infer stiffness and rigidity from images. Stiffness can be measured experimentally, and until it is, it shouldn't be discussed.

Problems that are also noteworthy but probably don't impact the conclusions:

5) Fig. S1 should include an image of a control cell.

6) "Thrill" is not really used in this way in modern English. I think if the authors stick with "kink" it will be clearer.

7) In Fig. S2 I can't make out the satellite colonies.

8) Figure 4 is under-annotated. The 5-nm periodicity, the tapered ends, and the locations in the big images where the inset images were taken from - all these things

should be indicated.

9) The writing is generally very good needs some improvement to standard English in a lot of places throughout, and I recommend that the authors get a bit of help with this.

Reviewer #2 (Remarks to the Author):

Lartigue et al report on the changes in cell morphology in cell wall-less bacteria of the Mollicutes family of bacteria. Spiroplasma contains an intriguing arrangement of MreB filaments next to an extended filament formed by "fibril" protein. This arrangement has been postulated to mediate motion of the helical Spiroplasmas by locally changing the helical handedness of cells. The authors have undertaken the interesting approach of expressing the array of five MreB isoforms and of fibril in Mycoplasma cells, which lack helical cell shape as well as pitch-driven motion. The authors show that a single MreB isoform is sufficient to induce helical cell shape, and dependent on which isoform is used, even helical dynamics. Expression of fibril in addition to MreB5 increases the efficiency of helical motion, and the complete array appears to mimic Spiroplasma motility.

The study suggests that not only helical cell shape can be achieved by intrinsic activity of MreB proteins, but even changes in helicity, which can drive cell movement. It confirms an earlier study in Current Biology that MreB5 is a major driver in this aspect in Spiroplasma, and extends the finding to show that it can also change cell shape in a bacterium lacking MreB proteins.

My main criticism is that the study is, in its current form, quite preliminary and descriptive, largely depending on subjective statements. I will be only quoting a few cases, but the results section is full of subjective e, non-quantitative data.

- line 117: "Control cells were pleomorphic as expected for Mcap, i.e. they were mostly short-rod shaped and coccoid, a few elongated ones (up to 8 microns) but none with a helical morphology." Please give numbers for classifications, because "a few" could be between 0.1 and 10%, i.e. ranging over orders of magnitude. Also, please show example fields of these cells – they are nowhere to be seen in the manuscript.

- line 121.." the cell morphology was heterogeneous and included helices, long, straight and soft filamentous cells, eventually branched (up to 80 microns), but also partially helical cells, and coccoid cells (Fig. 1A)." Please give hard numbers, how many cells were counted, how many showed which particular morphology. Please define "soft", to me, this applies to the physical aspect of rigidity, and not to shape. Fig. 1A shows a single cell – please show examples of fields of cells, at least in the supplement.

- line 175 "Mcapfib recombinants were characterized by a short-rod shape for most cells, but also by the presence of a few short helices having a helical pitch of $1.06 \pm 0.28 \mu\text{m}$ (Fig. 2)." Please quantify, what is meant by "a few". This applies to many more sites in the results part.

As a second point, the idea that expression of MreBs can induce helical cell shape and dynamic kinks in a heterologous cell is interesting, but not novel. The study would greatly benefit from more insight into how this is achieved. The authors could, for example, induce an ATPase deficient MreB5 variant in Mcap and analyse if kink-motility is based on ATPase activity. Further, how are SpMreBs anchored in the membrane? Is it an amphipathic helix like in E. coli, or a hydrophobic loop like in B. subtilis, or another mechanism? In the absence of membrane attachment, does MreB5 still achieve induction of helical cell shape? These would be experiment that really drive forward research on MreB proteins and their impact on cell shape maintenance.

Major points:

Line 184 "Also, some cell bodies showed certain flexibility, while others were characterized by a significant stiffness, possibly due to differences in amounts of cytoskeleton proteins." Please explain how flexibility and stiffness were measured, and substantiate what is meant by "certain".

Line 216 "However, only in the case of McapmreB1-5-fib, a bending of the rigid zone following a flexible point was observed." Please elaborate what is meant, the meaning is cryptic to me.

Fig. 4, Cryo-EM studies look interesting, but are difficult to digest for the non-specialist. In line 283, it is stated "The latter organization mimicked the Spiroplasma one, and differences observed between cells were likely due to differences from one cell to another in protein amounts." Please show corresponding images for Spiroplasma, to support this important point. I have a hard time finding structures shown by the Baumeister group in the EM images of the authors, filamentous structures in Mcap are a lot less well defined as shown for Spiroplasma.

Line 311 "...conferred rigidity to cells when interacting with large membrane..." see above, please explain how this was measured.

Line 317 ".." in particular in some tubular protrusions observed for some cells (Fig 4J)." Excessive non-quantitative data

Line 338 "induced disorganized cellular movements, 338 resembling tremors,..." please define what is meant by "tremors"

General points:

Some cells seem to show larger bulges or protrusions and to my eye look disrupted (Fig. 1, 2). Have the authors tried a live/dead stain, or membrane dye to determine if the amorphous cells with bulges are still viable? Lack of viability might explain lack of helical motion. Did the authors also capture this by CryoEM?

- Line 196-204: I have a hard time to clearly identify the propagation of membrane deformations the authors describe.

-

- Regarding kink movement - in the supplemented movies it appears that the cells are freely turning and moving quite quickly through the medium, is it possible to gently adhere the cells for microscopy, for example on agarose pads (poly-l-lysine likely would not work) to determine clearly that the described kink motion is not simply due to the free turning of the cells?

-

- Line 259-260: "However kink-based motility was not observed with the gene combination mreB5-fib." Is there an example movie or time lapse of this, for comparison to the actual described kink based motility? As mentioned it is difficult for me to discern between random movement of a flexible, helical cell, and actual directly driven kink propagation based on the images and movies provided.

- Judging by the proteomics in table 1 it is difficult to tell if indeed all 5 MreBs are expressed in the MreB 1-5-fib transformants. Have the authors tried to express MreBs 2-4 individually, and with the fibril respectively? Similar to what was done for MreBs 1 and 5. This would certainly give more insight into the postulated functions of the proteins and substantiate this analysis.

Response to Reviewers Comments

Nature Communications manuscript NCOMMS-21-47425

Our responses to the Reviewers' comments are in blue.

We would like to thank both reviewers for their careful and expert examination of our manuscript. We are very grateful to the reviewers for their constructive comments.

Please find below our responses to their comments. All changes are highlighted in the revised submitted version of our manuscript.

Reviewer #1 (Remarks to the Author):

The manuscript is based on a fascinating idea - using gene expression in an unrelated but highly plastic model organism, *Mycoplasma capricolum* to identify a minimal set of cytoskeletal or cytoskeleton-associated components that confers generation of *Spiroplasma citri*'s characteristic helical shape and unusual motility, providing a system to probe the contribution of each component to the *spiroplasma* phenotype. There are some considerable successes in this work. Using a set of five MreB genes and the gene for the fibril protein, the researchers most assuredly achieved the *spiroplasma*'s shape, as well as the characteristic propagation of the helix kink along the cell's long axis. They also had some partial success at probing the roles of each of the MreB isoforms. These cells did not achieve swimming through liquid, leading the authors to propose that other proteins are required for that process. The methodology is sufficiently well-described to allow reproducibility, and with perhaps one exception (see point #1) is appropriate. There are some elements of the manuscript that are confusing, and some of the authors' data lead me to believe that alternative interpretations of their results might better explain the phenomena the researchers observe.

We thank reviewer 1 for his kind comments on our work and his (explicitly) positive evaluation of the paper, and have addressed his comments below.

The biggest problems:

1) LC-MS data suggest that in different transformants, the amounts of each MreB isoform differ, and in most instances some isoforms are undetectable. Unfortunately, the absence of a transformant in which all isoforms plus fibril are produced, together with the fact that those that are produced are in wildly varying ratios, makes it quite difficult to establish a clear relationship between the presence of these proteins and the observed phenotypes. All things considered, rather than invoking additional factors required for swimming, it seems to me that the most straightforward explanation for the absence of swimming is that the MreB isoforms aren't all there in their normal ratios, so the structure that's assembled isn't normal. According to this, what's really missing is factors that generate and maintain the normal stoichiometry of these proteins in *S. citri*. Has that been characterized?

We totally agree with reviewer 1's thoughts and conclusion that:

- (1) A Mcap transformant in which all isoforms plus fibril are produced, was not observed
- (2) Amounts of each MreB isoform in the different Mcap transformants differ
- (3) And thus, it makes it difficult to establish a clear relationship between the presence of these proteins and the observed phenotypes

We agree that this part of the manuscript had to be re-structured to make it clear that the presented proteomics was not informative enough to get a clear correlation between the observed phenotypes and the expression of each specific MreBs. The primary goal of our manuscript was to define the minimal set of genes required to confer helicity and motility to *Mycoplasma* cells. For this objective, proteomics provided elements of information, and notably that MreB5 was expressed as a major protein in all *Mycoplasma* transformants showing kink-propagation. This observation led to the choice to conduct further experiments using transformants expressing MreB5 only or together with the fibril

protein. The phenotype of MreB5 transformants was compared to that of transformants expressing MreB1 that was also found as a major protein in *Spiroplasma* and in transformants.

We also agree with the reviewer's view that the assembly of the cytoskeleton in *Mycoplasma* transformants does not result in a functional structure in most transformants and this may be one possible reason why helical cells cannot move forward in liquid medium. Therefore, it is likely that a specific stoichiometry of the different MreBs is required for the cytoskeleton fibers to assemble into a functional structure, as discussed in the initial version of the manuscript lines 468-473. This was our justification for inserting genes under their native promoter and not using strong promoters when we started the study. Factors responsible for regulation of gene expression level in Mollicutes species are not well characterized, and although *Mycoplasma capricolum* and *Spiroplasma citri* are phylogenetically close among Mollicutes, specific regulators may be lacking (or differ for their affinity for different promoter sequences) in *Mycoplasma*. We therefore agree that one possible reason for the lack of directionality observed for the helical transformants showing kink propagation is the lack of some regulators maintaining the required stoichiometry of MreBs. Knowing the relative amounts of the different MreB isoforms allowing *Spiroplasma* motility would help getting some more clues in favor of this hypothesis.

However, this is not the sole possibility to explain the lack of directionality: proteomics being a global approach providing information on the protein content of a whole population, it is therefore possible that the correct stoichiometry is respected in mycoplasmas showing kink propagation but lacking ability to move forward in liquids. In this case, minor proteins present in a minor fraction of the cell population may not be detected. We have thus to envisage that the lack of other cell factors (polarized morphology, lack of attachment of the fibrils to the cell poles or of the dumbbell structure) may be responsible for the lack of directionality.

Actions taken: First, **new LC-MS/MS analyses** have been performed in order to better define the correct relative amounts of MreBs needed for *Spiroplasma* motility. We chose to analyse the MreB content in *S. citri* at different time points of growth (Supplementary Table 2) to have a better knowledge of possible variations of expression in *Spiroplasma*. These amounts have then been compared to those observed in *Mycoplasma* transformants. Proteomics for the most interesting clones have been performed on new cultures for the revised version on our manuscript to be able to correlate morphology, motility and protein content. Normalized abundances of proteins have been calculated on the basis of the relative amount of unique peptides. These results argue in favor of Reviewer 1 preferred hypothesis.

Second, the proteomics result section has been rewritten (lines 223-262) and hypotheses explaining the lack of directionality in Mcap transformants has been rephrased. As suggested by Reviewer 1, the first hypothesis has been emphasized (lines 484-491), while the other ones have been more succinctly discussed (lines 495-506). This part includes the hypothesis proposed by Reviewer 1 in his/her comment 3) below.

2) I'm confused by the use of the word "motility." Sometimes it seems to refer to swimming motion through the media. Sometimes it seems to refer to kink propagation. Sometimes I'm not really sure what it means. In this case, I would call movement of the whole cell "motility," and propagation of the kink "kink propagation."

The reviewer's comment is correct.

Action taken: First, in this new version, we have clearly defined the term "motility" in the introduction section (motility=movement of a whole cell through the medium) (lines 113-117). Second, we have checked and replaced with the correct wordings all expressions/sentences used that include the word "motility" in the text to avoid any ambiguity (lines 122, 175, 217, 277, 282, 349, 466, 476).

3) Relatedly, there is a significant logical flaw in the way the authors describe the relationship between kink propagation and swimming motility. Surely, in prior literature it has been hypothesized that kink "propagation is responsible for the movement of the bacteria," as the authors state on line 75 and

reiterate on line 200. However, the authors' own results clearly dismiss that model, as they have achieved kink propagation but not motility. This is an important point that they have not emphasized; it isn't actually clear that it's been considered.

We thank Reviewer 1 for this suggestion and agree with him/her.

Action taken: We have added the hypothesis that kink propagation frequency may not be sufficient to confer motility in *Spiroplasma* and emphasize this point in the discussion's section (lines 496-501).

4) I don't think it's appropriate to infer stiffness and rigidity from images. Stiffness can be measured experimentally, and until it is, it shouldn't be discussed.

Indeed, we agree that the use of the terms « stiffness » and « rigidity » may lead to confusions. Here the physical parameters corresponding to the cell stiffness were not measured. Stiffness referred to the physical aspect of the cells that were not deformed upon fluid movements (straight cells).

Action taken: as suggested by Reviewer 1, stiffness (and rigidity) has been removed from the text, and the removal of this idea did not impact the main conclusions of this work. We clarified that we hypothesize that an increase in cell stiffness could explain the lack of kinks in transformants with *mreb5* and *fib* genes: lines 366-369.

Problems that are also noteworthy but probably don't impact the conclusions:

5) Fig. S1 should include an image of a control cell.

Action taken: Reviewer probably meant Figure 1. A picture showing control cells (Mcap carrying an empty vector) has been added to this figure, as suggested by the reviewer. Larger fields of cells are also included as Supplementary Figure 2.

6) "Thrill" is not really used in this way in modern English. I think if the authors stick with "kink" it will be clearer.

Here again, we apologize for the vocabulary problems encountered by the reviewers.

Action taken : (1) These semantic mistakes (synonyms and polysemy) have been corrected. (2) As mentioned below, the manuscript has been sent to a professional English language editing service.

7) In Fig. S2 I can't make out the satellite colonies.

The quality of the picture is probably not optimal to show these satellites.

Action taken: A new culture on agar-containing medium has been performed, and a new photography is now provided (Supplementary Figure 4). Arrows point to satellites.

8) Figure 4 is under-annotated. The 5-nm periodicity, the tapered ends, and the locations in the big images where the inset images were taken from - all these things should be indicated.

Action taken: The required annotations have been added on Figure 4 following the reviewer's suggestion. Figure 4 is now completed by Supplementary Figure 8 regarding the periodicity observed within the structures.

9) The writing is generally very good needs some improvement to standard English in a lot of places throughout, and I recommend that the authors get a bit of help with this.

Action taken: The revised version of the manuscript has been sent to a professional English language editing service, AJE (American Journal Expert). A copy of the editing certificate is available as a separate file.

Reviewer #2 (Remarks to the Author):

Lartigue et al report on the changes in cell morphology in cell wall-less bacteria of the Mollicutes family of bacteria. *Spiroplasma* contains an intriguing arrangement of MreB filaments next to an extended filament formed by “fibril” protein. This arrangement has been postulated to mediate motion of the helical *Spiroplasmas* by locally changing the helical handedness of cells. The authors have undertaken the interesting approach of expressing the array of five MreB isoforms and of fibril in *Mycoplasma* cells, which lack helical cell shape as well as pitch-driven motion. The authors show that a single MreB isoform is sufficient to induce helical cell shape, and dependent on which isoform is used, even helical dynamics. Expression of fibril in addition to MreB5 increases the efficiency of helical motion, and the complete array appears to mimic *Spiroplasma* motility.

The study suggests that not only helical cell shape can be achieved by intrinsic activity of MreB proteins, but even changes in helicity, which can drive cell movement. It confirms an earlier study in *Current Biology* that MreB5 is a major driver in this aspect in *Spiroplasma*, and extends the finding to show that it can also change cell shape in a bacterium lacking MreB proteins.

We thank the reviewer for these positive comments.

To complete Reviewer 2 comments, the paper published previously in *Current Biology* described the complementation of a natural mutant strain (ASP-1) that was only missing the MreB5 protein. The mutant is not helical but displays a rod-shape morphology and the full *Spiroplasma citri* genome. Here, the starting cells are *Mycoplasma capricolum* that are spherical and devoid of cytoskeleton in a genetic background that is only phylogenetically related to *Spiroplasma*. Therefore, the present study is much more ambitious in reconstituting the full cell shape and motility displayed by *spiroplasmas*. Moreover, it allows defining a minimal set of genes capable of maintaining a helical shape in bacteria, which was never done before.

My main criticism is that the study is, in its current form, quite preliminary and descriptive, largely depending on subjective statements. I will be only quoting a few cases, but the results section is full of subjective e, non-quantitative data.

We thought that our results were demonstrative enough without requiring the quantification of all aspects. We thank the reviewer for this comment and we modified the manuscript accordingly. Changes made are highlighted in yellow. Hard numbers have been added when it was possible to quantify.

- line 117: “Control cells were pleomorphic as expected for Mcap, i.e. they were mostly short-rod shaped and coccoid, a few elongated ones (up to 8 microns) but none with a helical morphology.” Please give numbers for classifications, because “a few” could be between 0.1 and 10%, i.e. ranging over orders of magnitude. Also, please show example fields of these cells – they are nowhere to be seen in the manuscript.

Initially we thought that it was rather difficult to illustrate this heterogeneity because it is necessary to scan several fields of observation to be seen, but we finally succeeded in answering the reviewer’s comment.

Action taken: Quantitative data have been added to this part dedicated to the description of cell morphology in the *Mycoplasma* control population (mean length of control cells, line 129; dispersion of the cell length values is given in Figure 1C). A figure with fields of cells for all transformants is now provided as Supplementary Figure 2.

- line 121..” the cell morphology was heterogeneous and included helices, long, straight and soft filamentous cells, eventually branched (up to 80 microns), but also partially helical cells, and coccoid cells (Fig. 1A).” Please give hard numbers, how many cells were counted, how many showed which particular morphology. Please define “soft”, to me, this applies to the physical aspect of rigidity, and not to shape. Fig. 1A shows a single cell – please show examples of fields of cells, at least in the supplement.

Action taken: Fields of cells have been added (Fig. 1a and Supplementary Figure 2) for the transformants having the whole set of mreB genes and fib. Hard numbers are now provided. Percentages of helical, nonhelical and branched cells in the population of cells is given lines 133-134. Number of counted elongated cells (150) was added line 132.

The word “Soft” was used to mean deformable cells, this was corrected in the revised version of the manuscript (lines 151-153).

- line 175 “Mcapfib recombinants were characterized by a short-rod shape for most cells, but also by the presence of a few short helices having a helical pitch of $1.06 \pm 0.28 \mu\text{m}$ (Fig. 2).” Please quantify, what is meant by “a few”. This applies to many more sites in the results part.

Action taken: Quantitative data related to cell numbers have been added throughout the entire manuscript. In the case of Mcap^{fib} transformants, less than 1% of the cells were helical and this information was added line 184.

As a second point, the idea that expression of MreBs can induce helical cell shape and dynamic kinks in a heterologous cell is interesting, but not novel. The study would greatly benefit from more insight into how this is achieved. The authors could, for example, induce an ATPase deficient MreB5 variant in Mcap and analyse if kink-motility is based on ATPase activity. Further, how are SpMreBs anchored in the membrane? Is it an amphipathic helix like in *E. coli*, or a hydrophobic loop like in *B. subtilis*, or another mechanism? In the absence of membrane attachment, does MreB5 still achieve induction of helical cell shape? These would be experiment that really drive forward research on MreB proteins and their impact on cell shape maintenance.

Regarding the novelty of the present work:

We agree with Reviewer 2 that the link between cell shape and MreB was already shown in other bacterial native and heterologous systems. However, the *Spiroplasma* model differs from other bacterial systems because of the lack of a cell wall and because of the presence of 5 MreB paralogs. It is therefore not possible to directly infer knowledge on other bacterial MreBs to all *Spiroplasma* MreB isoforms. In the opposite, the diversity of MreBs in *Spiroplasma* can help studies aiming at identifying sequence specificities responsible for the role of these proteins in bacterial cell shape. As indicated above, we have previously demonstrated that MreB5 acts as a major determinant of cell helicity in *Spiroplasma* because it restores helicity in a non-helical variant lacking a functional *mreb5* gene but expressing all 4 other functional *mreb* and *fib* genes (Harne et al., 2020, Curr. Biol.). However, we could not demonstrate that MreB5 was polymerized in vivo, due the presence of all 4 other MreBs in the non-helical variant, and for the same reason we could not prove that MreB5 had a direct role in helicity. In the present work we show that MreB5 can polymerize in vivo in the *Mycoplasma* host which is very close to *Spiroplasma*, and we demonstrate its direct role in helicity, because in this system it was possible to express MreB5 only. In addition, we show that MreB1 can also induce cell helicity, but not with the right helical pitch.

To our knowledge, the link between MreBs and swimming motility has not been made before. This is an important point in our study as we demonstrate that a single MreB is able to induce the propagation of membrane deformations. In addition, we demonstrate that kinks are formed in the absence of the protein fibril, which was previously proposed to be responsible for *Spiroplasma* motility.

We agree with Reviewer 2 that the numerous experiments that he is proposing will provide valuable clues about the mechanism of shape maintenance in Mollicutes and will allow to identify functional contribution of each MreBs. However, we consider that these points should be considered as part of another study together with other mechanistic aspects (source of energy, type of interaction with the membrane, interaction between MreBs...). In the present publication, our goal was to demonstrate that the mycoplasma cells constitute an outstanding model for understanding the mechanism of shape control and motility, that the MreBs were involved in motility and that only one MreB made it possible to obtain the propagation of kinks, which allows significant advances in the understanding of the functions of bacterial MreBs. Besides the fact that understanding the entire mechanism was not at the center of this work, practical constraints prevent all of the suggested experiments from being carried out in a reasonable time period. Indeed, as stated in the publication, the introduction of MreBs induces division problems, reducing transformation efficiency. Mycoplasmas and spiroplasmas being fastidious bacteria, the time required to obtain transformants observable in culture is a minimum of 6 months. Obtaining the transformants described in this work took 2 years. The obtention of the desired transformants is never guaranteed, and it is also a bet on the implication of such or such MreB sequence part. We are thus currently undertaking this functional study but it will be the subject of a future manuscript.

Major points:

Line 184 “Also, some cell bodies showed certain flexibility, while others were characterized by a significant stiffness, possibly due to differences in amounts of cytoskeleton proteins. “ Please explain how flexibility and stiffness were measured, and substantiate what is meant by “certain”.

We agree with Reviewer 2 and apologize for the confusion here. We are referring to the stiffness in the aspect of the cell, the fact that it does not deform from the movement of liquids, and not from the stiffness of the cell body as a physical variable.

Actions taken: The text was revised to avoid this confusion; this comment was also made by reviewer 1. The words soft, stiffness (and rigidity) has been removed from the result section, and the removal of this idea did not impact the main conclusions of this work. In the discussion, we clarified that we hypothesize that an increase in cell stiffness could explain the lack of kinks in transformants with mreb5 and fib genes, and that this assumption will have to be assessed in future studies: lines 364-367.

Line 216 “However, only in the case of McapmreB1-5-fib, a bending of the rigid zone following a flexible point was observed.” Please elaborate what is meant, the meaning is cryptic to me.

We fully understand that our description may not be clear enough.

Action taken: We have illustrated this type of movement by adding a movie as supplementary information (in Supplementary Movie 2).

Fig. 4, Cryo-EM studies look interesting, but are difficult to digest for the non-specialist. In line 283, it is stated “The latter organization mimicked the Spiroplasma one, and differences observed between cells were likely due to differences from one cell to another in protein amounts.” Please show corresponding images for Spiroplasma, to support this important point. I have a hard time finding structures shown by the Baumeister group in the EM images of the authors, filamentous structures in Mcap are a lot less well defined as shown for Spiroplasma.

The misunderstanding here likely comes from a lack of clarity in our description. It seems that Reviewer 2 understood that variations in the amounts of proteins led to changes in the structure of the cytoskeleton in the spiroplasma. We have no evidence to support this. Here our hypothesis is that in transforming mycoplasmas, the different morphologies observed (helical, non-helical, long filaments...) could be attributed to different levels of expression of heterologous proteins. The

difference between the studies from Baumeister's group and ours is very likely due to the fact that cryotomography was not used here.

Action taken: We have clarified this point and make it clear that this is only a hypothesis and place this clarification in the discussion section (lines 490-492).

Line 311 "...conferred rigidity to cells when interacting with large membrane..." see above, please explain how this was measured.

As explained above, the stiffness as a physical parameter has not been measured. Here we meant that the presence of many cytoskeleton structures in a single cell triggered loss of helicity leading to straight filamentous cells.

Action taken: the sentence has been removed. In the discussion section, lines 416-417, the idea has been clarified as follows: 'Notably, bundles of thick filaments were associated with a straight cell morphology, indicating the likely requirement of a thinner filament for helicity.'

Line 317 ".." in particular in some tubular protrusions observed for some cells (Fig 4J)." Excessive non-quantitative data

We agree with the reviewer's comment

Action taken: The corresponding sentence has been removed, as we considered that it was not possible to quantify the number of cells showing protrusions using cryoEM. Accordingly, Figure 4j has also been removed.

Line 338 "induced disorganized cellular movements, 338 resembling tremors,..." please define what is meant by "tremors"

As discussed above in response to the reviewer 1, we realize that we needed to use a more specific and defined vocabulary describing the different aspects of motility.

Action taken: This point was clarified. As described above, all types of movements observed in the study were better defined. As suggested by Reviewer 1, the term motility was defined in the Introduction section (Lines 113-117). For a better clarity, the text is now more focused on the capacity to propagate kinks associated with a helicity switch.

General points:

Some cells seem to show larger bulges or protrusions and to my eye look disrupted (Fig. 1 , 2). Have the authors tried a live/dead stain, or membrane dye to determine if the amorphous cells with bulges are still viable? Lack of viability might explain lack of helical motion. Did the authors also capture this by CryoEM?

The apparent bulges on these photographs are short branches, which can appear as disrupted cell parts depending on the orientation of the cell in the medium. Such structures are frequently observed in *Mycoplasma* cells grown under non-optimal conditions (Seto & Miyata, 1998, J. Bacteriol. 180 : 256–264 ; Seto & Miyata, 1999, J. Bacteriol. 181:6073-80; Lartigue et al., 2007, Science 317:632-638). Cells with branching constituted a large cell subpopulation in our experiments (hard numbers are given Lines 133-134). The increase in cell length through passages in axenic medium indicates that these cells are able to grow (and thus are alive), but have a problem in septation and in localization of sites of initiation of new cell poles.

Action taken: LIVE/DEAD staining was applied in order to answer this question raised by Reviewer 2. Results of this assay are now provided in Supplementary Figure 3. Branched cells are viable.

The three following points raised by Reviewer 2 are linked and our answers are as follow:

– Line 196-204: I have a hard time to clearly identify the propagation of membrane deformations the authors describe.

We agree that the propagation of membrane deformations is not easy to follow on images.

Action taken: Typical kinks were annotated on Fig. 3. For McapmreB5 and Mcap^{fib}, membrane propagation was associated with loss of helicity, rendering more difficult the illustration of such movements as the cell moves in the 3 dimensions. To help the reader, we added models built using Blender 3D starting from the movies from which the images were extracted; In addition, as suggested by Reviewer 2 (see the two next points), the movie showing a cell without propagating membrane deformations (Supplementary Movie 2) will certainly help the reader to see the cell movements.

– Regarding kink movement – in the supplemented movies it appears that the cells are freely turning and moving quite quickly through the medium, is it possible to gently adhere the cells for microscopy, for example on agarose pads (poly-L-lysine likely would not work) to determine clearly that the described kink motion is not simply due to the free turning of the cells?

We agree with Reviewer 2 that the small size of Mollicutes is responsible for their high sensitivity to Brownian motion and to fluid flow. Unfortunately, we did not succeed in attaching *Mycoplasma* cells on the glass slide using poly-L-lysine or agarose. We also failed in slowing down Brownian motion by adding methylcellulose. All these treatments triggered cell deformations. The addition of agarose impeded the correct diffusion of light in darkfield microscopy.

Action taken: The movies provided in this new version of the manuscript show cells which are not moving quickly through the medium, thanks to the sealing of the microscope slides in these new experiments. We have also recorded a movie of a cell showing no membrane deformation (see next point) and add it in Supplementary Movie 2.

– Line 259-260: “However kink-based motility was not observed with the gene combination mreB5-fib.” Is there an example movie or time lapse of this, for comparison to the actual described kink based motility? As mentioned it is difficult for me to discern between random movement of a flexible, helical cell, and actual directly driven kink propagation based on the images and movies provided.

Action taken: A movie and a time lapse of a non-kinking cell has been added for comparison.

– Judging by the proteomics in table 1 it is difficult to tell if indeed all 5 MreBs are expressed in the MreB 1-5-fib transformants. Have the authors tried to express MreBs 2-4 individually, and with the fibril respectively? Similar to what was done for MreBs 1 and 5. This would certainly give more insight into the postulated functions of the proteins and substantiate this analysis.

By performing new LC-MS/MS analyses, we confirm that all MreBs are either not produced or produced at very low levels. As explained in point 1 raised by Reviewer 1, the assembly of the cytoskeleton in *Mycoplasma* transformants is probably not fully functional in most transformants and this could explain why helical cells cannot move forward in liquids. We fully agree with Reviewer 2 on the interest of these other constructs using our *Mycoplasma* model in studies aiming at deciphering the motility mechanism in *Spiroplasma*. Our aim was here to focus on the demonstration that MreB5 alone could induce the kinks and conferred a helicity similar to those of *Spiroplasma citri*. The *Mycoplasma* model will be used for further studies in order to better understand the role of each MreB isoform through functional genomics studies. However, as explained above, production of new constructs is time consuming (more than 6 months) and not guaranteed, and will therefore be the subject of future studies.

Reviewer #1 (Remarks to the Author):

All my concerns have been adequately addressed in the revision.

Reviewer #2 (Remarks to the Author):

Main point:

Fig. 1A and B: panel A is way too small, cells look like "dirt", it is basically impossible to see their actual shape. I would a) increase the size of the panel as much as possible and show a smaller field of cells (enough to see a sufficient number of cells), b) show one control cell in panel B at the same magnification as present cells in the panels. It is important to really grasp the change from the perspective of discerning a spherical cell.

Title: Just as a suggestion: MreB rules all results, so why not include it in the title? "MreB - driven Turing of" This is really a matter of taste.

Otherwise, all my points raised have been dealt with in an ok manner.

Response to Reviewers Comments

Nature Communications manuscript NCOMMS-21-47425A

Our responses to the Reviewers' comments are in blue.

We would like to thank both reviewers for their positive comments and suggestions. Please find below our responses to their comments. Corresponding changes are highlighted in the revised submitted version of our manuscript.

Reviewer #1 (Remarks to the Author):

All my concerns have been adequately addressed in the revision

Reviewer #2 (Remarks to the Author):

Main point:

Fig. 1A and B: panel A is way too small, cells look like "dirt", it is basically impossible to see their actual shape. I would a) increase the size of the panel as much as possible and show a smaller field of cells (enough to see a sufficient number of cells), b) show one control cell in panel B at the same magnification as present cells in the panels. It is important to really grasp the change from the perspective of discerning a spherical cell.

Fig. 1a and 1b have been changed according to the Reviewer's suggestion. Fig 1a shows a smaller field of cells, allowing the size of the images to be increased by a factor 2.5. This facilitates the visualisation of the morphological aspects of the cells. Fig. 1b now includes a representative image of the short rod-shape morphology of most *Mcap^{control}* cells.

Title: Just as a suggestion: MreB rules all results, so why not include it in the title? "MreB - driven Turing of" This is really a matter of taste.

We agree that it is pertinent to include the fact that the introduction of cytoskeleton elements is responsible for the changes observed. We changed the title following the Editor suggestion into 'Cytoskeletal components can turn wall-less spherical bacteria into kinking helices'.

Otherwise, all my points raised have been dealt with in an ok manner.